# Glypican-1 drives unconventional secretion of fibroblast growth factor 2

Carola Sparn[1], Eleni Dimou[1†], Annalena Meyer[1†], Roberto Saleppico[1†], Sabine Wegehingel[1], Matthias Gerstner[1], Severina Klaus[1], Helge Ewers[2], Walter Nickel[1]*

[1]Heidelberg University, Biochemistry Center, Heidelberg, Germany; [2]Institut für Chemie und Biochemie, Freie Universität Berlin, Berlin, Germany

**Abstract** Fibroblast growth factor 2 (FGF2) is a tumor cell survival factor that is transported into the extracellular space by an unconventional secretory mechanism. Cell surface heparan sulfate proteoglycans are known to play an essential role in this process. Unexpectedly, we found that among the diverse subclasses consisting of syndecans, perlecans, glypicans, and others, Glypican-1 (GPC1) is the principle and rate-limiting factor that drives unconventional secretion of FGF2. By contrast, we demonstrate GPC1 to be dispensable for FGF2 signaling into cells. We provide first insights into the structural basis for GPC1-dependent FGF2 secretion, identifying disaccharides with N-linked sulfate groups to be enriched in the heparan sulfate chains of GPC1 to which FGF2 binds with high affinity. Our findings have broad implications for the role of GPC1 as a key molecule in tumor progression.

## Editor's evaluation

FGF2 moves directly from the cytoplasm through the plasma membrane in a reaction driven by its subsequent high affinity binding to cell surface heparan sulfate proteoglycans. This study, surprisingly, identifies Glypican-1 as the principal proteoglycan involved, possibly involving a unique tri-sulfated disaccharide binding site in close proximity to the cell surface. Thus, Glypican-1 is new component in the pathway of unconventional secretion of FGF2.

**\*For correspondence:**
walter.nickel@bzh.uni-heidelberg.de

[†]These authors contributed equally to this work

**Competing interest:** The authors declare that no competing interests exist.

## Introduction

Proteoglycans are components of the extracellular matrix and play essential roles in the storage and protection of growth factors, chemokines, and morphogens that bind to the glycosaminoglycan chains of proteoglycans on cell surfaces (*Schlessinger et al., 1995*; *Ori et al., 2011*). These post-translational modifications are polymerized into unbranched chains of repetitive disaccharide building blocks. They can be classified into four categories defined by (i) heparan sulfates (HSPGs), (ii) chondroitin sulfates, (iii) keratan sulfates, and (iv) hyaluronic acid. Heparan sulfates are characterized by about 20–300 negatively charged residues with almost infinite structural modifications such as epimerization and sulfation patterns that are dynamically processed by enzymes resulting in variations between tissues, developmental stages, and the type of core protein they are attached to (*Turnbull et al., 2001*). Different classes of HSPGs also differ in terms of how they are anchored to membranes in that glypicans (GPCs) contain a glycosylphosphatidylinositol (GPI) anchor whereas syndecans (SDCs) carry transmembrane spans. Based on their differential modes of membrane association, GPCs and SDCs partition into liquid ordered and disordered domains, respectively, providing a structural basis for distinct roles in growth factor signaling (*Gutiérrez and Brandan, 2010*). As part of their functions to modulate cell growth and differentiation, various kinds of proteoglycans are known to play key roles

in tumorigenesis and cancer progression. Their expression patterns along with the structural aspects discussed above and their potential release from cell surfaces play crucial roles in the coordination of their biological functions. These may differ between different kinds of proteoglycans depending on tissue types, developmental stages of tumors, and different tumor microenvironments (*Blackhall et al., 2001*; *De Pasquale and Pavone, 2020*; ). Proteoglycans are also known to play key roles in the development of chemoresistances making them suitable drug targets for anti-cancer therapies (*Lanzi et al., 2017*).

Among the proteins that bind to the alternating negatively charged disaccharide units of heparan sulfate chains in HSPGs is fibroblast growth factor 2 (FGF2) (*Lindahl et al., 1999*; *Murphy et al., 2004*), a pro-angiogenic factor involved in cell proliferation and differentiation during development. In addition, under pathophysiological conditions, FGF2 has a strong impact on tumor-induced angiogenesis triggering the formation of new blood vessels to provide the large demands of malignant cancers for nutrients and oxygen (*Carmeliet, 2000*; *Akl et al., 2015*; *Akl et al., 2016*). FGF2 also plays a critical role as a tumor cell survival factor blocking programmed cell death through both autocrine and paracrine signaling (*Okada-Ban et al., 2000*; *Akl et al., 2016*; *Traer et al., 2016*). Therefore, blocking the biological functions of FGF2 by either limiting its secretion into the extracellular space or inhibiting FGF2 signaling into cells are suitable strategies in anti-cancer treatments (*Akl et al., 2016*; *Pallotta and Nickel, 2020*).

While the majority of extracellular proteins contain N-terminal signal peptides for ER-Golgi-dependent protein secretion (*Palade, 1975*; *Rothman, 1994*; *Rothman and Wieland, 1996*; *Schekman and Orci, 1996*), FGF2 lacks a signal peptide and thus does not have access to the ER/Golgi-dependent secretory pathway (*Nickel, 2005*; *Nickel, 2007*). Instead, FGF2 is secreted into the extracellular space by an unconventional mechanism of protein secretion (*Rabouille, 2017*; *Dimou and Nickel, 2018*; *Pallotta and Nickel, 2020*). Various kinds of such pathways have been identified that were collectively termed 'unconventional protein secretion' (UPS) (*Malhotra, 2013*; *Rabouille, 2017*; *Dimou and Nickel, 2018*; *Pallotta and Nickel, 2020*). The mechanism by which FGF2 is transported into the extracellular space is based on direct protein translocation across the plasma membrane (UPS Type I) (*Schäfer et al., 2004*; *Zehe et al., 2006*; *Rabouille, 2017*; *Steringer et al., 2017*; *Dimou and Nickel, 2018*; *Dimou et al., 2019*; *Pallotta and Nickel, 2020*).

All molecular components known to date to play a role in unconventional secretion of FGF2 are physically associated with the plasma membrane. These factors include the Na,K-ATPase (*Zacherl et al., 2015*), Tec kinase which is recruited to the inner leaflet via binding to $PI(3,4,5)P_3$ (*Ebert et al., 2010*; *Steringer et al., 2012*; *La Venuta et al., 2016*), and $PI(4,5)P_2$ (*Temmerman et al., 2008*; *Temmerman and Nickel, 2009*; *Nickel, 2011*), the most abundant phosphoinositide at the inner leaflet of the plasma membrane (*Di Paolo and De Camilli, 2006*). In the above-mentioned studies, FGF2 has been demonstrated to engage in direct physical interactions with all three of these components with the Na,K-ATPase being the first contact for FGF2 at the inner plasma membrane leaflet (*Legrand et al., 2020*). Through subsequent interactions with $PI(4,5)P_2$ mediated by a cluster of basic amino acids on the molecular surface of FGF2 (K127, R128, and K133 [*Temmerman et al., 2008*; *Müller et al., 2015*; *Steringer et al., 2017*]), the core mechanism of FGF2 membrane translocation is triggered. This process involves membrane insertion of FGF2 oligomers (*Steringer et al., 2012*; *Steringer et al., 2017*; *Steringer and Nickel, 2018*) whose biogenesis depends on two surface cysteines in FGF2 that drive oligomerization through the formation of intermolecular disulfide bridges (*Müller et al., 2015*; *Steringer et al., 2017*; *Dimou and Nickel, 2018*). Membrane-inserted FGF2 oligomers are accommodated within a lipidic membrane pore with a toroidal architecture (*Steringer et al., 2012*; *Müller et al., 2015*; *Steringer and Nickel, 2018*). This conclusion was derived from several independent observations including simultaneous membrane passage of fluorescent tracers and transbilayer diffusion of membrane lipids triggered by $PI(4,5)P_2$-dependent FGF2 oligomerization and membrane insertion (*Steringer et al., 2012*; *Steringer and Nickel, 2018*). In further support of this, diacylglycerol, a cone-shaped lipid that interferes with membrane curvature stabilized by $PI(4,5)P_2$, was found to inhibit membrane insertion of FGF2 oligomers (*Steringer et al., 2012*; *Steringer and Nickel, 2018*), a typical phenomenon for toroidal membrane pores (*Gilbert et al., 2014*). Based upon these findings, the role of $PI(4,5)P_2$ in unconventional secretion of FGF2 has been proposed to be threefold with (i) mediating recruitment of FGF2 at the plasma membrane, (ii) orienting FGF2 molecules at the inner leaflet to drive oligomerization, and (iii) stabilizing local curvature to allow for

a toroidal membrane structure surrounding membrane-inserted FGF2 oligomers that are accommodated within a hydrophilic environment (*Dimou and Nickel, 2018*; *Steringer and Nickel, 2018*).

As discussed above, membrane-inserted FGF2 oligomers have been proposed to act as key intermediates in FGF2 membrane translocation based on an assembly/disassembly mechanism driving directional transport of FGF2 across the plasma membrane (*Dimou and Nickel, 2018*; *Steringer and Nickel, 2018*). This process depends on membrane-proximal HSPGs on cell surfaces that capture and disassemble FGF2 translocation intermediates, thereby mediating the final step of FGF2 transport into the extracellular space (*Zehe et al., 2006*; *Nickel, 2007*; *Nickel and Seedorf, 2008*; *Nickel and Rabouille, 2009*). A critical property of heparan sulfates for this function is their ability to outcompete PI(4,5)P$_2$ with regard to physical interactions toward FGF2. These are mutually exclusive with heparan sulfates having an about 100-fold higher affinity for FGF2 compared to PI(4,5)P$_2$ (*Steringer et al., 2017*). FGF2 on cell surfaces undergoes intercellular spreading by direct cell-cell contacts, probably mediated by direct exchange between heparan sulfate chains that are physically associated with opposing cell surfaces (*Zehe et al., 2006*). Thus, during the lifetime of an FGF2 molecule, the role of HSPGs is threefold with (i) mediating the final step of FGF2 secretion (*Zehe et al., 2006*; *Nickel, 2007*), (ii) protecting FGF2 on cell surfaces against degradation and denaturation (*Nugent and Iozzo, 2000*), and (iii) mediating FGF2 signaling as part of ternary complexes containing FGF2, heparan sulfate chains, and FGF high-affinity receptors (*Presta et al., 2005*; *Ribatti et al., 2007*; *Belov and Mohammadi, 2013*).

In conclusion, based upon sequential interactions of FGF2 with PI(4,5)P$_2$ at the inner plasma membrane leaflet and, following the formation of membrane-spanning FGF2 oligomers, interactions with heparan sulfates on cell surfaces, the proposed mechanism of FGF2 membrane translocation offers a molecular basis for directional FGF2 transport into the extracellular space. It has recently been confirmed in a fully reconstituted system using giant unilamellar vesicles (*Steringer et al., 2017*) and is consistent with earlier observations demonstrating that membrane translocation depends on a fully folded state of FGF2 that permits PI(4,5)P$_2$-dependent FGF2 oligomerization and interactions with heparan sulfate chains (*Backhaus et al., 2004*; *Torrado et al., 2009*). Furthermore, PI(4,5)P$_2$- and heparan-sulfate-dependent translocation of FGF2 across the plasma membrane has also been visualized in living cells using single molecule TIRF microscopy. These studies revealed the real-time kinetics of this process with an average time interval for FGF2 membrane translocation of about 200 ms (*Dimou et al., 2019*; *Pallotta and Nickel, 2020*).

In the current study, we made the intriguing and unexpected discovery that HSPGs of different kinds cannot serve equally well in capturing FGF2 on cell surfaces as the final step of its unconventional secretory mechanism. Instead, using a proteome-wide BioID screen, we identified GPC1 as the principle HSPG driving this process. Even though HeLa cell lines lacking GPC1 were found to contain normal amounts of total glycosaminoglycans, FGF2 secretion was severely impaired. This phenotype could be reversed by re-expression of GPC1. By contrast, GPC5, the second family member of GPCs expressed in HeLa cells, failed to rescue the GPC1 knockout as did SDC4, an HSPG from the SDC family. Following the purification of various ectodomains from GPCs and SDC4, the quantification of the binding kinetics revealed a strong preference of FGF2 for GPC1. These findings were corroborated by a strongly increased binding of recombinant FGF2-GFP to the surface of cells overexpressing GPC1. Furthermore, using purified components, FGF2 was found to bind to the heparan sulfate chains of GPC1 with much higher affinity compared to those of GPC5 and SDC4. Based on analytical methods, we found disaccharide units enriched in the heparan sulfate chains of GPC1 that are known to play a role in recruiting FGF2. The strong FGF2 binding efficiency toward GPC1 could therefore be based on a unique arrangement of the corresponding hexasaccharide FGF2 binding units forming clusters with high avidity in the glycosaminoglycan chains of GPC1. As opposed to its critical role driving efficient secretion of FGF2, we found GPC1 to be dispensable for FGF2 signaling. Our studies reveal a novel and unexpected functional specialization of an HSPG with major implications for the prominent role of GPC1 in tumor progression.

## Results

### GPC1 and FGF2 are in proximity to cell surfaces

To unveil so far unidentified proteins that are in proximity to FGF2 at any time of its lifetime in intact cells, we conducted a proteome-wide BioID screen. A HeLa S3 cell line was generated expressing a fusion protein of FGF2 and the promiscuous biotin ligase BirA (*Roux et al., 2012*). In a control cell line, a myc-tagged form of BirA was expressed. Both constructs were stably integrated into the genomes of the corresponding HeLa S3 cell lines expressing these fusion proteins in a doxycycline-dependent manner. Following 48 hr of incubation of doxycycline-induced cells in the presence or absence of biotin, a Western analysis was performed to visualize biotinylated proteins under the conditions indicated (*Figure 1A*). This analysis revealed distinct patterns of biotinylated proteins when cell lines expressing FGF2-BirA were compared with those expressing BirA alone (*Figure 1A*, lane 5 versus lane 7). By contrast, a post-lysis addition of biotin did not affect the patterns of biotinylated proteins, irrespective of whether conditions with or without biotin in the culture medium were compared (*Figure 1A*). These observations indicate that the vast majority of biotinylated proteins was generated in viable cells before lysis.

To focus on proteins in proximity to FGF2 that are not localized to the nucleus, a fractionation protocol was established to remove nuclei from all other membranes and cytosolic components. As shown in *Figure 1B* and described in detail in Materials and methods, a fraction containing both α-tubulin (as a cytosolic marker) and the Na,K-ATPase (as a plasma membrane marker) could be generated that is devoid of nuclear markers such as histone-H3 and NCBP1. Based on the procedures described in *Figure 1A and B*, fractions with nuclear factors being removed were prepared from both FGF2-BirA- and myc-tagged BirA-expressing cells. Biotinylated proteins were pulled down with streptavidin beads, subjected to SDS-PAGE followed by a Western analysis (*Figure 1C*). The biotinylated fraction of proteins was subjected to a mass spectrometry analysis to identify all proteins that were in proximity to FGF2 at the level of intact cells (Fingerprints Proteomics Facility at Dundee University, Scotland).

A comparative protein quantification between FGF2-BirA- and myc-BirA-containing fractions was conducted based on peptide intensities. Based on three replicates, for each hit, the differences in peptide intensities between FGF2-BirA and myc-tagged BirA lysates (log$_2$, fold change) were plotted against the negative log$_{10}$ p-value (*Figure 1D*). The resulting volcano plot identified proteins in the upper right corner that were more abundant in the FGF2-BirA fraction in a statistically significant manner. This analysis revealed known interaction partners of FGF2 such as API5 (*Noh et al., 2014*; *Bong et al., 2020*). In addition, FGF2 itself was identified likely due to its ability to oligomerize at the inner leaflet of the plasma membrane.

The strongest hit of this screen was GPC1, a GPI-anchored HSPG associated with cell surfaces (*Figure 1D*). This was a surprising finding for several reasons. First, BioID screens typically return intracellular proteins as hits since ATP is needed to activate biotin to be transferred by BirA to target proteins. The half-life of the BirA-biotinyl-5′-AMP complex of the BirA* R188G mutant enzyme used in our BioID screens is in the range of 5 s (*Kwon and Beckett, 2000*; *Oostdyk et al., 2019*). By comparison, FGF2-GFP has been shown to translocate across the plasma membrane within an average time interval of 200 ms (*Dimou et al., 2019*). Therefore, the observed biotinylation of cell surface GPC1 mediated by FGF2-BirA* is consistent with the kinetic data on the stability of the BirA*-biotinyl-5′-AMP complex and the time interval it takes for FGF2 to translocate from the inner to the outer leaflet of the plasma membrane. Second, no other cell surface HSPGs were found in proximity to FGF2. These observations were taken as evidence that GPC1 may represent an HSPG that is intimately linked to sites of FGF2 membrane translocation with a specialized function in unconventional secretion of FGF2.

### GPC1 is a rate-limiting component of the FGF2 secretion machinery

Following the identification of GPC1 as a cell surface HSPGs in cellular proximity to FGF2 (*Figure 1*), we engineered cell lines with knockouts of GPC1 and GPC5, the two family members expressed in HeLa S3 cells (*Figure 2—figure supplement 1*). This included single and double knockouts along with cell lines from each knockout background being stably modified to re-express either GPC1 or GPC5 for rescue experiments (*Figure 2—figure supplement 1*; panel A). Further cell lines were generated expressing each member of the glypican family (GPC1–6) in a GPC1 knockout background (*Figure 2—figure supplement 1*; panel B). Finally, we engineered GPC1 knockout cell lines in which a GPC1 version with a transmembrane domain (instead of the natural GPI anchor) or SDC4, an HSPG

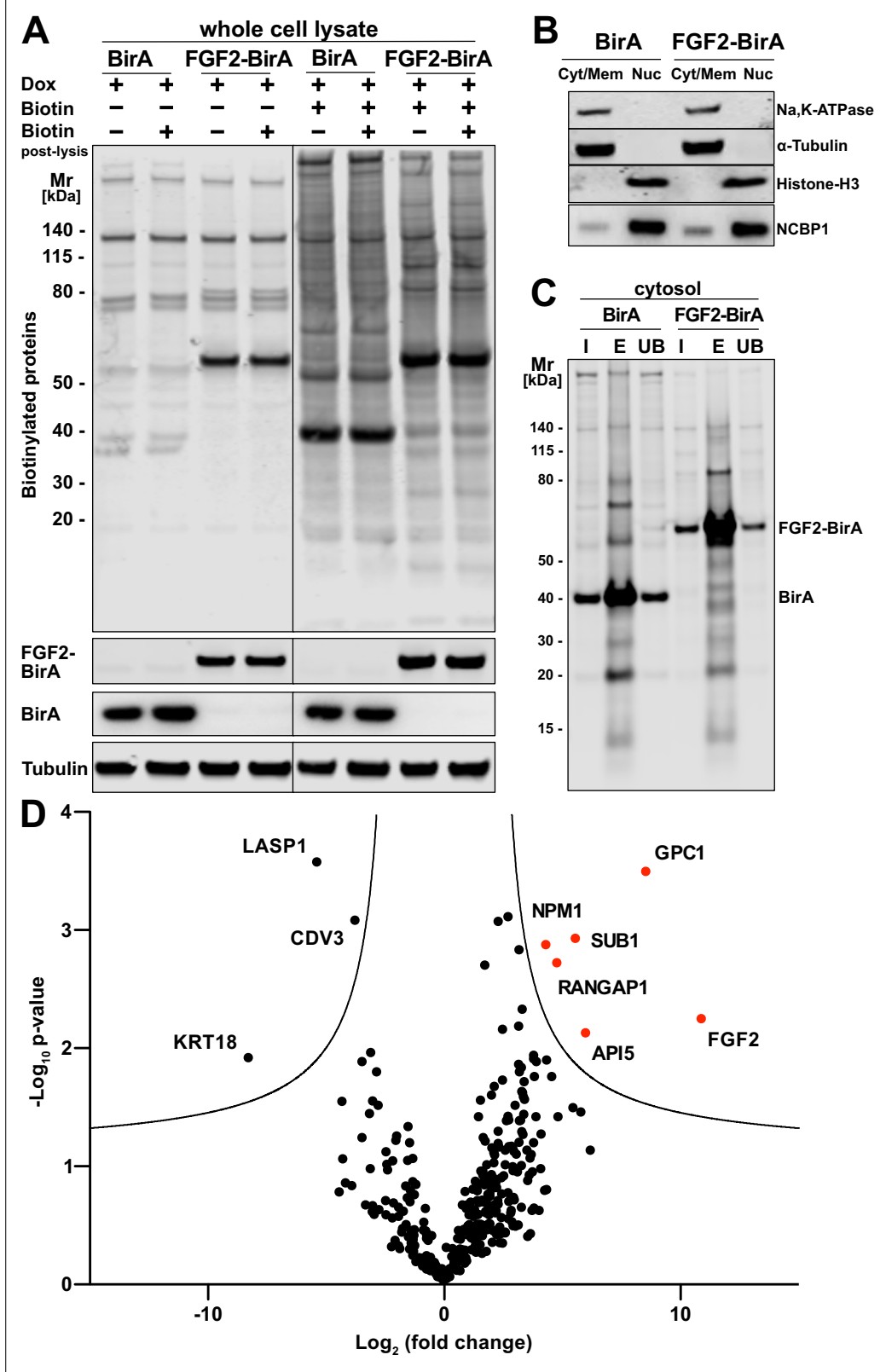

**Figure 1.** Glypican-1 (GPC1) and fibroblast growth factor 2 (FGF2) are in spatial proximity to a cellular context. (**A**) HeLa S3 cells stably expressing either FGF2-BirA or myc-tagged BirA (control) in a doxycycline-dependent manner were cultured as detailed in the Materials and methods section. Whole cell lysates generated from the experimental conditions indicated were subjected to a Western blot analysis. Biotinylated proteins were identified

*Figure 1 continued on next page*

*Figure 1 continued*

with fluorescent streptavidin. The expression of the fusion proteins was tested with antibodies directed against FGF2 (for FGF2-BirA) or the myc epitope (for BirA). In all samples, tubulin was used as a loading control. (**B**) HeLa S3 cells expressing FGF2-BirA or myc-tagged BirA were fractionated into nuclei (Nuc) and cellular membranes plus cytosol (Cyt/Mem) as described in the Materials and methods section. The fractionation was controlled by markers for the plasma membrane (Na,K-ATPase), the cytosol (α-tubulin), and nuclear proteins (histone-H3 and NCBP1). (**C**) Large-scale preparations of nuclei-free fractions from both FGF2-BirA and myc-tagged BirA expressing cell lines containing biotinylated target proteins. Based on the Cyt/Mem fractionation shown in panel B, all biotinylated proteins were isolated using streptavidin beads. Following elution (lane 'E'), all regions except those containing the BirA fusion proteins were extracted and subjected to a quantitative mass spectrometry analysis shown in panel D. For details see Materials and methods. (**D**) Biotinylated proteins identified by mass spectrometry and visualized by a Volcano plot indicating hits based on their relative abundance in FGF2-BirA versus myc-tagged BirA-expressing cells. The quantification was based on peptide intensities expressed as 'x-fold change' ($\log_2$; FGF2-BirA/myc-BirA). The experiment was performed in three replicates from which p-values ($-\log_{10}$) were calculated (unpaired t-test, two-sided). For further details, see Materials and methods.

The online version of this article includes the following source data for figure 1:

**Source data 1.** Raw data of the BioID screening procedure identifying proteins in proximity to fibroblast growth factor 2 (FGF2) in intact cells.

from the SDC family (characterized by membrane anchors based on transmembrane spans) were expressed (*Figure 2—figure supplement 1*; panel C).

The engineered HeLa cell lines described in *Figure 2—figure supplement 1* (panel A) were analyzed for their ability to secrete FGF2 (*Figure 2*). A well-established biotinylation assay was used to quantify FGF2-GFP on cell surfaces (*Figure 2A*; *Engling et al., 2002*; *Seelenmeyer et al., 2005*; *Zehe et al., 2006*; *Müller et al., 2015*; *La Venuta et al., 2016*; *Legrand et al., 2020*). A representative Western analysis used for quantification is shown in *Figure 2C*. To validate the cell surface biotinylation experiments, we also quantified FGF2 on cell surfaces using a well-established flow cytometry assay (*Figure 2B*; *Engling et al., 2002*; *Backhaus et al., 2004*; *Stegmayer et al., 2005*; *Temmerman et al., 2008*; *Ebert et al., 2010*; *Zacherl et al., 2015*; *Ran et al., 2013*). For both read-outs, all experimental conditions were normalized against HeLa wild-type cells (*Figure 2A and B*, dotted lines) and differences were evaluated for statistical significance. These experiments revealed a strong decrease in FGF2 secretion efficiency when GPC1 was absent. By contrast, a knockout of GPC5 did not impact this process. Consistently, a double knockout of GPC1 and GPC5 did not further intensify the FGF2 secretion phenotype observed in cells in which only GPC1 was knocked out. In all cell lines described, overexpression of GPC1 did not only rescue the knockout of the endogenous GPC1 gene but rather increased the efficiency of FGF2 secretion to levels well above HeLa wild-type cells. As shown in *Figure 2—figure supplement 2* and in the *Videos 1–3*, endocytosis of FGF2-GFP along a time course of 60 min could neither be detected in HeLa S3 wild-type, GPC1 knockout, nor in GPC1 knockout cells overexpressing GPC1. Therefore, the observed differences in the cell surface levels of FGF2 (*Figure 2*) represent true FGF2 secretion phenotypes that were not compromised by increased rates of endocytosis in GPC1 knockout cells.

We further analyzed the cell lines described above with regard to their cell surface capacities to recruit FGF2-GFP (*Figure 3A*), their total contents of glycosaminoglycan chains (*Figure 3B*), and their total amounts of heparan sulfate chains (*Figure 3C*). Using a flow cytometry assay to quantify binding of recombinant FGF2-GFP to cell surfaces, we found that cells lacking GPC1 display slightly reduced binding capacities for FGF2-GFP. By contrast, a GPC5 knockout did not affect FGF2-GFP binding to cell surfaces. Strikingly, all types of cell lines overexpressing GPC1 were characterized by significantly increased binding capacities for FGF2-GFP. Again, GPC5 overexpression did not result in increased binding of FGF2-GFP to cell surfaces (*Figure 3A*). These observations were made despite the fact that there were no significant differences in the total amounts of both glycosaminoglycan chains (*Figure 3B*) and heparan sulfate chains (*Figure 3C*) in all engineered cell lines analyzed in comparison to HeLa wild-type cells. The combined findings documented in *Figures 2 and 3* suggest that, among the various kinds of HSPGs expressed in mammalian cells, GPC1 is the principal component that drives efficient secretion of FGF2. The data further indicate that GPC1 is the rate-limiting factor among the components of the FGF2 secretion machinery that appears to have strong binding

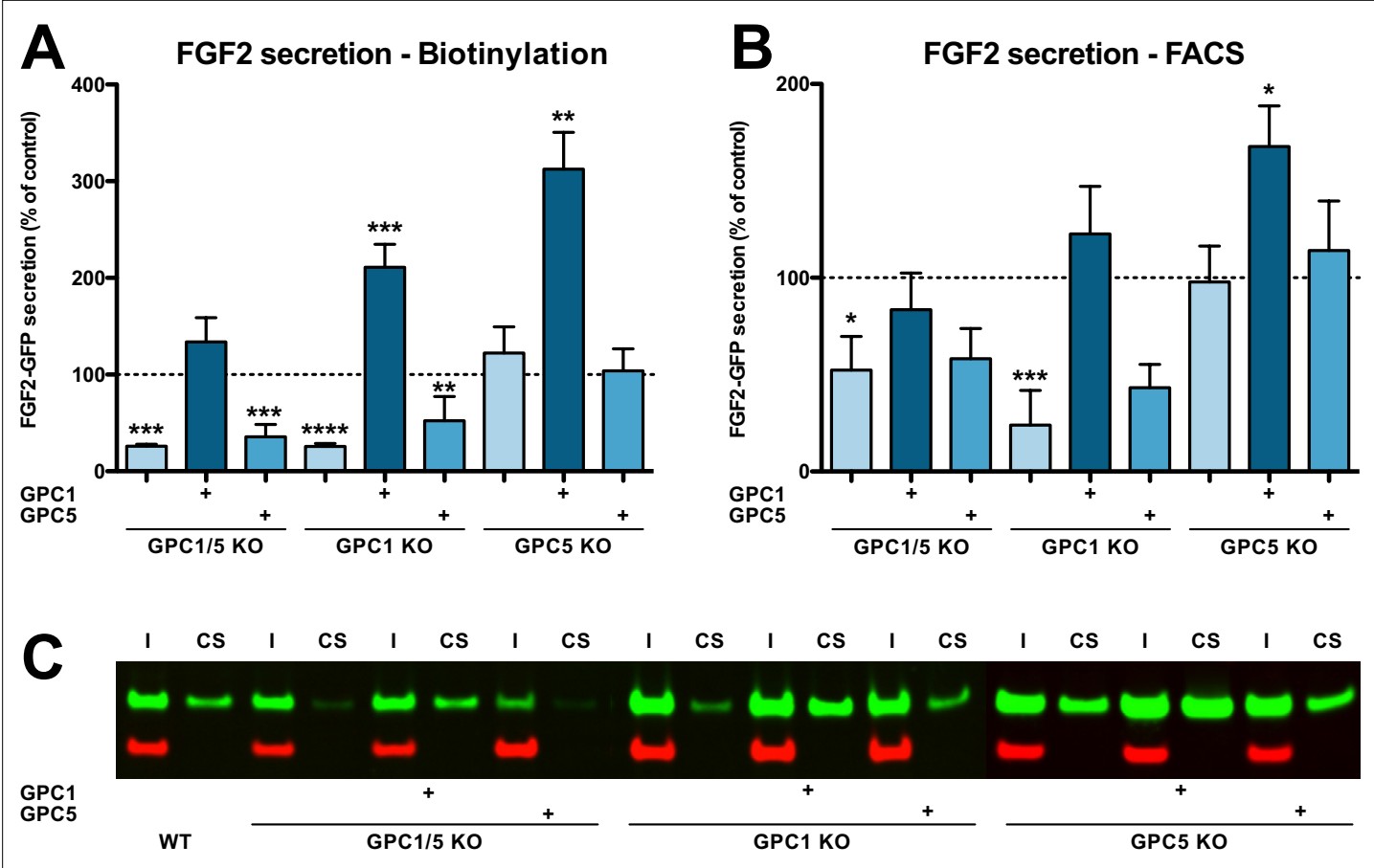

**Figure 2.** Efficient secretion of fibroblast growth factor 2 (FGF2) to cell surfaces depends on Glypican-1 (GPC1). (**A**) Quantitative analysis of FGF2 secretion under the experimental conditions indicated measured by cell surface biotinylation. Standard deviations are shown (n = 4). (**B**) Quantitative analysis of FGF2 secretion under the experimental conditions indicated measured by analytical flow cytometry. Standard deviations are shown (n = 5). (**C**) Representative example of a cell surface biotinylation experiment used for the quantitative analysis and statistics shown in panel A (I = input; CS = cell surface). Statistical analyses were based on a two-tailed t-test (*, p ≤0.05; **, p ≤0.01, and ***, p ≤0.001). For details, see Materials and methods.

The online version of this article includes the following source data and figure supplement(s) for figure 2:

**Source data 1.** Raw data of the cell surface biotinylation and flow cytometry experiments quantifying fibroblast growth factor 2 (FGF2) secretion under the conditions indicated.

**Figure supplement 1.** Characterization of engineered HeLa S3 cells used to quantify fibroblast growth factor 2 (FGF2) secretion efficiencies.

**Figure supplement 2.** Visualization of endocytosis of fluorescent transferrin versus fibroblast growth factor 2 (FGF2)-GFP in HeLa S3 cells comparing wild-type, Glypican-1 (GPC1) knockout and GPC1 knockout plus GPC1 overexpression conditions.

capabilities toward FGF2 since GPC1 overexpression causes increased binding of FGF2-GFP to cell surfaces (*Figure 3A*) without affecting the total amounts of cellular heparan sulfate chains (*Figure 3C*).

Based on the cell lines described and characterized in *Figure 2*, *Figure 2—figure supplement 1*, we tested whether overexpression of other GPC family members can rescue FGF2 secretion in the context of a GPC1 knockout (*Figure 4*). Of note, the GPC family can be divided into two subclasses, GPC1/2/4/6 and GPC3/5. As shown in *Figure 4A and C*, like GPC5, GPC3 was incapable of promoting efficient FGF2 secretion in the absence of GPC1. By contrast, all GPCs belonging to the GPC1 subfamily rescued FGF2 secretion in a GPC1 knockout background. While GPC2 and GPC4 did so at the level of HeLa wild-type cells, GPC6 overexpression increased the efficiency of FGF2 secretion above HeLa wild-type levels. Nevertheless, GPC1 overexpression was found to represent the strongest stimulator of FGF2 secretion. As shown in *Figure 4B*, these findings were closely reflecting the ability of the various GPC family members to increase the cell surface binding capacities for FGF2-GFP.

We also tested whether other HSPGs such as SDCs can support efficient secretion of FGF2 in a GPC1 knockout background. As shown in *Figure 4D and E*, SDC4 overexpression neither rescued

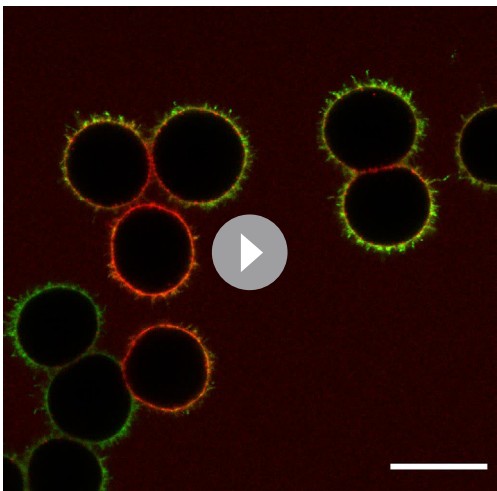

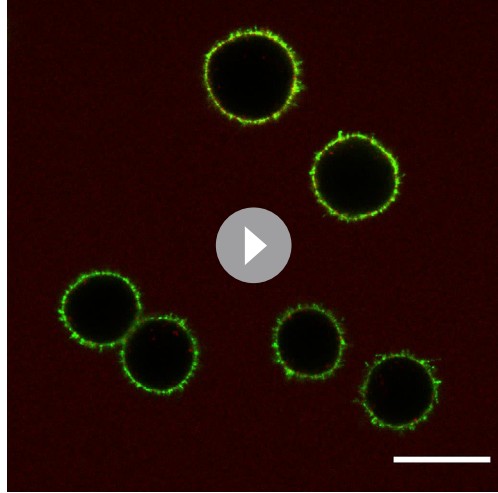

**Video 1.** Endocytosis of recombinant fibroblast growth factor 2 (FGF2)-GFP (5 µg/ml, in green) and Transferrin-Alexa Fluor546 (25 µg/ml, in red) was visualized in HeLa S3 wild-type cells. One frame corresponds to 1 min. The data shown are representative for three independent experiments. Scale bar = 20 µm.

https://elifesciences.org/articles/75545/figures#video1

**Video 3.** Endocytosis of recombinant fibroblast growth factor 2 (FGF2)-GFP (5 µg/ml, in green) and Transferrin-Alexa Fluor546 (25 µg/ml, in red) was visualized in HeLa S3 knockout cells overexpressing Glypican-1 (GPC1). One frame corresponds to 1 min. The data shown are representative for three independent experiments. Scale bar = 20 µm.

https://elifesciences.org/articles/75545/figures#video3

FGF2 secretion nor did it affect the cell surface binding capacities for recombinant FGF2-GFP.

This observation was not due to the fact that SDC4 and GPC1 structurally differ in membrane attachment with SDC4 carrying a transmembrane span and GPC1 having a GPI anchor. This was particularly evident from the fact that an engineered version of GPC1 in which the GPI anchor was replaced by a membrane span ('GPC1 TM') was functional in GPC1 knockout cells, both with regard to supporting efficient secretion of FGF2 (*Figure 4D*) and cell surface binding capacities for recombinant FGF2-GFP (*Figure 4E*).

To assess the impact of GPC1 overexpression on the efficiency by which FGF2 is secreted from cells at different FGF2-GFP expression levels, we used an advanced TIRF assay with single molecule resolution (*Dimou et al., 2019*). These experiments were conducted in CHO cells that express FGF2-GFP in a doxycycline-dependent manner, reading out the number of FGF2-GFP particles on the cell surface of individual cells (*Figure 5*). Two conditions were chosen characterized by high (*Figure 5A*) and low (*Figure 5C*) expression levels of FGF2-GFP. When cells were analyzed expressing FGF2-GFP at high levels, the average number of FGF2-GFP particles on cell surfaces was increased by about 50% in a pool of GPC1-overexpressing cells relative to CHO wild-type cells (*Figure 5B*). This difference was even more pronounced in cells expressing low levels of FGF2-GFP resulting in a more than fourfold higher average number of FGF2-GFP particles on the cell surfaces of GPC1-overexpressing cells compared to wild-type cells (*Figure 5D*). Since the pool of GPC1-overexpressing cells was characterized by

**Video 2.** Endocytosis of recombinant fibroblast growth factor 2 (FGF2)-GFP (5 µg/ml, in green) and Transferrin-Alexa Fluor546 (25 µg/ml, in red) was visualized in HeLa S3 Glypican-1 (GPC1) knockout cells. One frame corresponds to 1 min. The data shown are representative for three independent experiments. Scale bar = 20 µm.

https://elifesciences.org/articles/75545/figures#video2

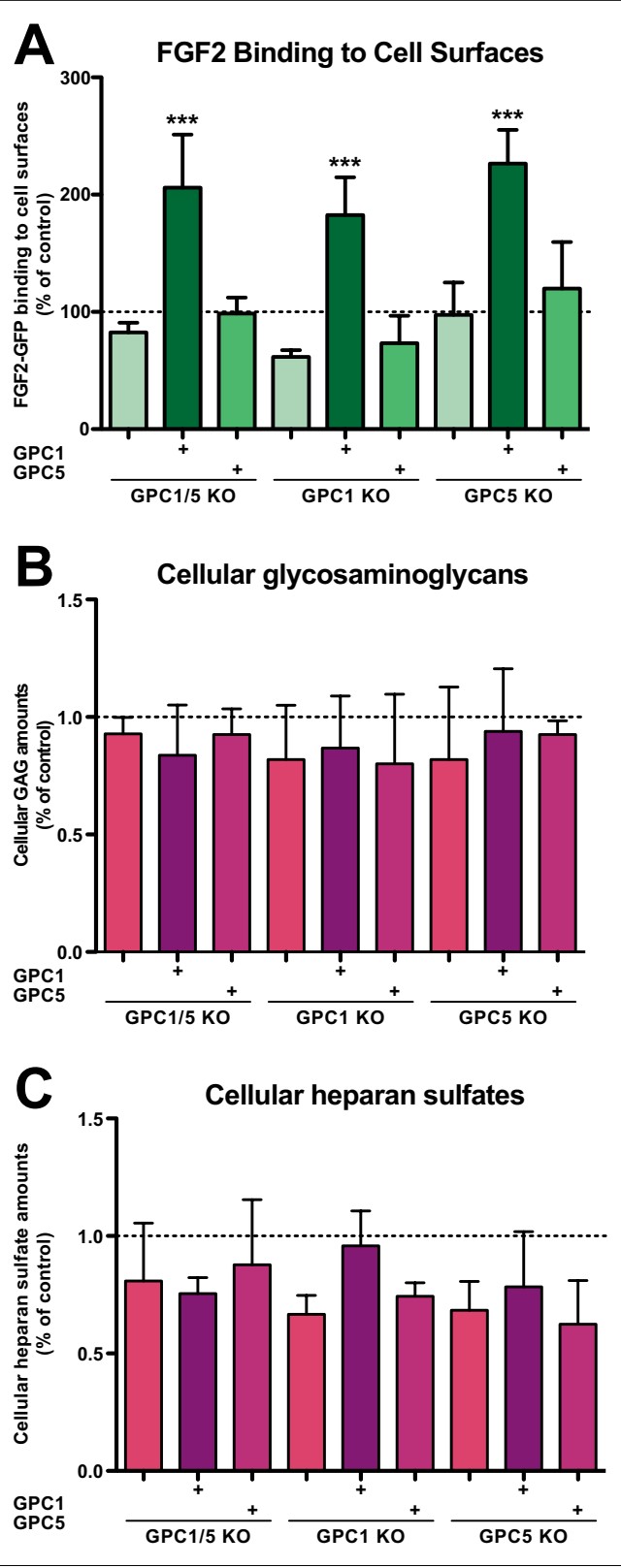

**Figure 3.** Fibroblast growth factor 2 (FGF2)-GFP binding to cell surfaces is increased in Glypican-1 (GPC1)-overexpressing cells. (**A**) Quantitative analysis of the FGF2-GFP binding capacity of cell surfaces under the experimental conditions indicated using flow cytometry. Standard deviations are shown (n = 5). Statistical significance was analyzed using a one-way ANOVA test combined with Tukey's post hoc test (*, p ≤ 0.05; **, p ≤

*Figure 3 continued on next page*

*Figure 3 continued*

0.01, and ***, p ≤ 0.001). (**B**) Quantification of the total amounts of GAG chains under the experimental conditions indicated. Standard deviations are shown (n = 4). Statistical significance was analyzed using a one-way ANOVA test combined with Tukey's post hoc test (*, p ≤ 0.05; **, p ≤ 0.01, and ***, p ≤ 0.001). (**C**) Quantification of the total amounts of heparan sulfate chains under the experimental conditions indicated. Standard deviations are shown (n = 3). Statistical significance was analyzed using a one-way ANOVA test combined with Tukey's post hoc test (*, p ≤ 0.05; **, p ≤ 0.01, and ***, p ≤ 0.001). For details, see Materials and methods.

The online version of this article includes the following source data for figure 3:

**Source data 1.** Raw data of experiments quantifying fibroblast growth factor 2 (FGF2) binding to cell surfaces as well as glycosaminoglycan and heparan sulfate contents of cells under the conditions indicated.

a range of GPC1 expression levels, a certain heterogeneity of this effect was observed. Strikingly, individual cells were observed that were characterized by a more than 20-fold increase of FGF2-GFP particles on their cell surface compared to the average number of FGF2-GFP particles on the cell surfaces of wild-type cells (*Figure 5D*). These findings suggest that GPC1 has an even stronger impact on this process when the amounts of FGF2-GFP being expressed are limiting.

The combined findings shown in *Figures 2–5* are in line with previous studies in which unconventional secretion of FGF2 has been shown to depend on the heparan sulfate chains of proteoglycans by using either cellular mutants that are incapable of attaching O-linked sugars to the core protein or by treatment of wild-type cells with chlorate, a condition that prevents the sulfation of the O-linked sugar chains of HSPGs (*Zehe et al., 2006*; *Dimou et al., 2019*). In the current study, we now identify GPC1 as the principal HSPG that drives the unconventionally secretory mechanism by which FGF2 is transported to the extracellular surfaces of cells.

## GPC1 and FGF2 form a strong pair of interaction partners

To study the interaction between the heparan sulfate chains from various kinds of GPCs and SDCs with FGF2 at the molecular level, we generated constructs of GPC1, GPC5, GPC6, and SDC4 to express and purify soluble variants of them from mammalian HEK cells (*Figure 6—figure supplement 1*; *Svensson et al., 2009*). In case of GPC1, to be used as a negative control, an additional variant form was generated that is defective regarding the addition of heparan sulfate chains (GPC1-ΔHS). As shown in *Figure 6—figure supplement 1*, all purified GPC family members were treated with heparinase III to demonstrate the presence of O-linked heparan sulfate chains. For SDC4, treatments with both heparinase III and chondroitin sulfate degrading enzyme were conducted to reveal the different types of O-glycosylation of this type of proteoglycan. In addition, recombinant FGF2 was purified to homogeneity (*Figure 6—figure supplement 1*).

To study physical interactions of FGF2 with the O-linked heparan sulfate chains of the proteoglycans indicated, we chose biolayer interferometry as read-out (*Figure 6*). Following biotinylation, all proteoglycans indicated were immobilized on BLI sensors. The sensors were then brought into contact with a range of FGF2 concentrations between 0.8 and 60 nM. This approach allowed for a quantitative comparison of the binding preferences of FGF2 to various kinds of heparan sulfate chains linked to different types of proteoglycans. It revealed a strong interaction of FGF2 with GPC1 that was detectable already at the lowest FGF2 concentration being used at 0.8 nM (*Figure 6A*). By contrast, even at the highest concentration of FGF2 (60 nM), GPC5 only showed weak interactions with FGF2 (*Figure 5C*) that were barely above the levels of the negative controls, GPC1-ΔHS (*Figure 6B*) or heparan sulfate binding mutants of FGF2 tested against GPC1 (*Figure 6—figure supplement 2*). Unlike GPC5, GPC6, a member of the GPC1 sub-family of GPCs that was capable of rescuing a GPC1 knockout (*Figure 4A and B*), displayed significant binding capabilities of FGF2, however, less efficiently compared to GPC1 (*Figure 6D*). Finally, similar to GPC5, SDC4, a member of the SDC family of proteoglycans, showed weak interactions with FGF2 at concentrations of up to 60 nM (*Figure 6E*).

As documented in *Figure 6—figure supplement 2*, beyond further controls using heparan sulfate binding mutants of FGF2, we found GPC1 to exhibit only weak or no interactions with other examples of growth factors or cytokines such as EGF and IFNγ. Similarly, examples for other extracellular proteins secreted by unconventional means such as galectin-1 and galectin-3 (*Rabouille, 2017*; *Dimou and Nickel, 2018*; *Pallotta and Nickel, 2020*) were observed not to be capable of interacting with GPC1 (*Figure 6—figure supplement 2*).

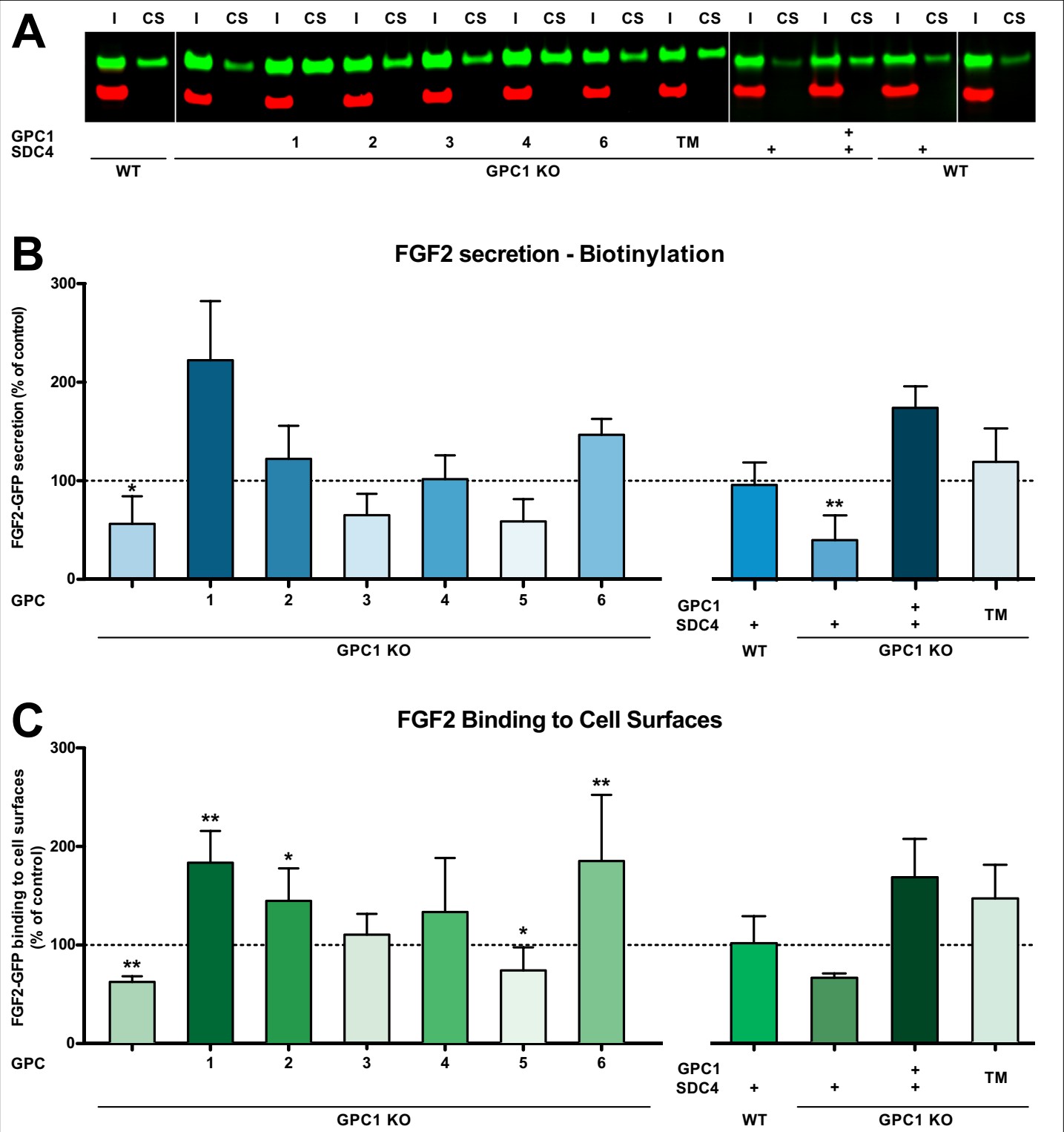

**Figure 4.** Glypican-1 (GPC1) is the principal heparan sulfate proteoglycan involved in unconventional secretion of fibroblast growth factor 2 (FGF2). (**A**) Representative example for the raw data of cell surface biotinylation experiments used to quantify and to statistically evaluate unconventional secretion of FGF2 under the conditions indicated (I = input; CS = cell surface). Standard deviations are shown (n = 5). (**B**) Quantitative comparison of all six GPC family members, GPC1 with a transmembrane anchor ('TM') and SDC4 (syndecan 4) with regard to their potential to drive FGF2 secretion upon overexpression in a GPC1 knockout background based on cell surface biotinylation experiments. Standard deviations are shown (n = 5). (**C**) Quantitative comparison of all six GPC family members, GPC1 with a transmembrane anchor ('TM') and SDC4 (syndecan 4) with regard to their potential to affect the

*Figure 4 continued on next page*

*Figure 4 continued*

cell surface binding capacities for FGF2-GFP. FGF2-GFP binding to cell surfaces was analyzed by flow cytometry. Standard deviations are shown (n = 4). Statistical analyses were based on a two-tailed t-test (*, p ≤0.05; **, p ≤0.01, and ***, p ≤0.001). For details, see Materials and methods.

The online version of this article includes the following source data for figure 4:

**Source data 1.** Raw data of the cell surface biotinylation and flow cytometry experiments quantifying fibroblast growth factor 2 (FGF2) secretion and FGF2 cell surface binding under the conditions indicated.

The combined findings shown in *Figure 6*, *Figure 6—figure supplement 1*, and *Figure 6—figure supplement 2* reveal a tight relationship between GPC1 and FGF2 that form a strong pair of interaction partners compared to other proteoglycans, growth factor, and cytokines including examples of other proteins secreted by unconventional means. They are consistent with the prominent role of GPC1 as the driver of the unconventional secretory mechanism of FGF2 as shown in *Figures 2–5*.

## The heparan sulfate chains of GPC1 are enriched in disaccharides known to be critical for FGF2 recruitment

To obtain insight into the molecular mechanism underlying the strong interaction between GPC1 and FGF2, we aimed at analyzing the disaccharide contents of the heparan sulfate chains of GPC1 in comparison to GPC5 and SDC4. Since there is no methodology available to sequence the disaccharide units of heparan sulfate chains, we treated the recombinant purified forms of GPC1, GPC5, and SDC4 (*Figure 6—figure supplement 1*) with a mixture of heparinase I, II, and III to convert their heparan sulfate chains into disaccharides. Using an established HPLC protocol (*Carnachan and Hinkley, 2017*; see Materials and methods for details), a total of 12 different heparan sulfate disaccharide standards with different sugar combinations and sulfation patterns (Iduron, UK) were analyzed for their retention times on an HPLC ion exchange column (*Figure 7—figure supplement 1*). These were compared with the retention times of the spectrum of dissacharide units released from GPC1, GPC5, and SDC4 upon treatments with heparinases (*Figure 7A*). The relative abundances of each of the identified disaccharides in the heparans sulfate chains of GPC1, GPC5, and SDC4, respectively, were quantified. The observed differences were tested for statistical significance (*Figure 7B*). This analysis revealed the enrichment of disaccharides in GPC1 over GPC5 that correspond to the disaccharide standards 1, 2, and 5. All of these disaccharides contain N-linked sulfates (*Figure 7—figure supplement 1*). The biggest difference between GPC1 and GPC5 was found to be the disaccharide standard 1 that represents a tri-sulfated disaccharide with two O-linked and one N-linked sulfate group. Intriguingly, when FGF2 was co-crystallized with synthetic heparin molecules, the binding site was found to contain three disaccharides of the type represented by the disaccharide standard 1 (*Raman et al., 2003*). By contrast, disaccharides lacking both O- and N-linked sulfations corresponding to the disaccharide standards 6 and 12 (*Figure 7—figure supplement 1*) were more abundant in GPC5 compared to GPC1 (*Figure 7B*). When the spectrum of disaccharides from GPC1 and GPC5 was compared with SDC4, most features were similar to GPC1 (disaccharide standards 1, 2, 6, and 12) while the abundance of the disaccharide corresponding to standard 5 was rather similar to GPC5 (*Figure 7A and B*). These findings suggest that, beyond the overall abundance of certain sulfated disaccharide units in heparan sulfate chains, their combination into trimers of sulfated disaccharides plays a key role in forming a high-affinity binding site for FGF2. The strong interaction of GPC1 with FGF2 (*Figure 6*) therefore indicates that GPC1 carries heparan sulfate units consisting of three disaccharides corresponding to standard 1 (*Figure 7—figure supplement 1*) as identified in structural in vitro studies (*Raman et al., 2003*). In addition, the heparan sulfate chains of GPC1 may contain multiple copies of these FGF2 hexasaccharide ligands in a clustered manner producing high avidity and, therefore, a strong apparent affinity toward FGF2. Based on our findings, such binding sites are likely to be less abundant in GPC5 and SDC4 resulting in weaker interactions with FGF2 (*Figure 6*). This, in turn, explains the predominant function of GPC1 in unconventional secretion of FGF2 as demonstrated in *Figures 2–5*.

## GPC1 is dispensable for FGF2-induced ERK1/2 signaling

To test as to whether GPC1 is not only the key driver of FGF2 secretion but also plays a role in FGF2 signal transduction, we analyzed the ability of recombinant FGF2 to initiate signal transduction

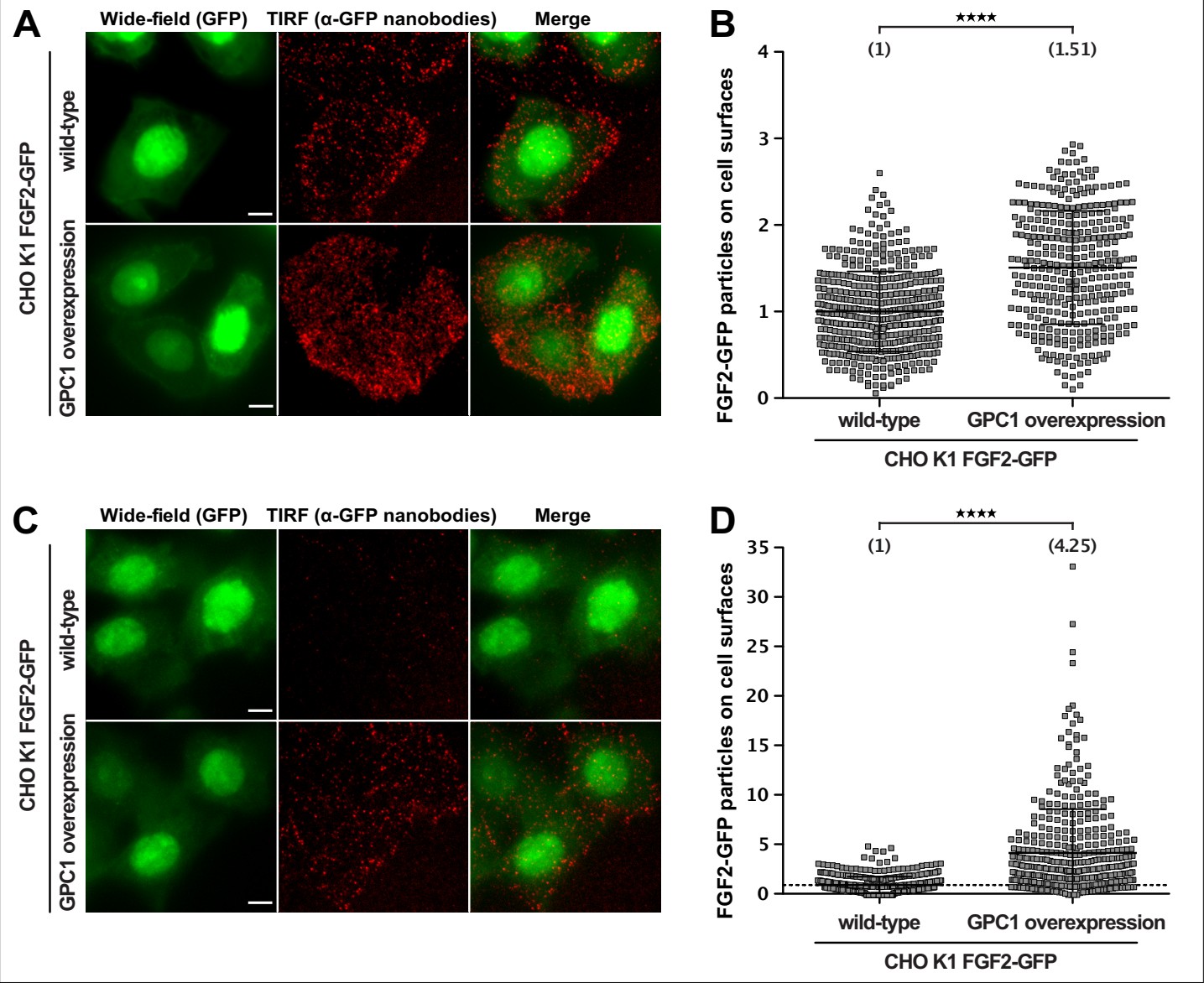

**Figure 5.** Glypican-1 (GPC1) overexpression results in increased fibroblast growth factor 2 (FGF2) secretion efficiencies. FGF2 secretion efficiencies in wild-type and GPC1-overexpressing cells were assessed by TIRF microscopy using anti-GFP nanobodies to detect single FGF2-GFP molecules on cell surfaces as described earlier (*Dimou et al., 2019*). For details, see Materials and methods. (**A**) Representative examples under experimental conditions at high FGF2-GFP expression levels. (**B**) Quantification and statistical analysis of experiments corresponding to the experimental conditions shown in panel A. (**C**) Representative examples under experimental conditions at low FGF2-GFP expression levels. EM Gain of the wide-field (GFP) was increased for this condition, in order to allow selection of cell area for subsequent quantification. (**D**) Quantification and statistical analysis of experiments corresponding to the experimental conditions shown in panel B. Data are shown as mean ± SD (n = 4) (panels B and D). The secretion efficiency of the wild-type cells was set to 1; in panel D, a dotted line was put at 1, to facilitate visualization. The statistical analysis was based on an unpaired t-test (****, p < 0.0001).

The online version of this article includes the following source data for figure 5:

**Source data 1.** Raw data of the cell-based TIRF experiments quantifying fibroblast growth factor 2 (FGF2) recruitment at the inner plasma membrane leaflet and FGF2 translocation to the outer plasma membrane leaflet under the conditions indicated.

in GPC1 knockout versus wild-type versus GPC1-overexpressing cells (*Figure 8*). As a read-out, we quantified ERK1/2 phosphorylation, an event that occurs downstream of FGF receptor activation. As a positive control, we treated the different cell types indicated with heparinases I, II, and III to degrade cell surface heparan sulfates down to their disaccharide subunits. The latter are incapable of forming ternary FGF signaling complexes on the surfaces of target cells consisting of FGF2, high-affinity FGF

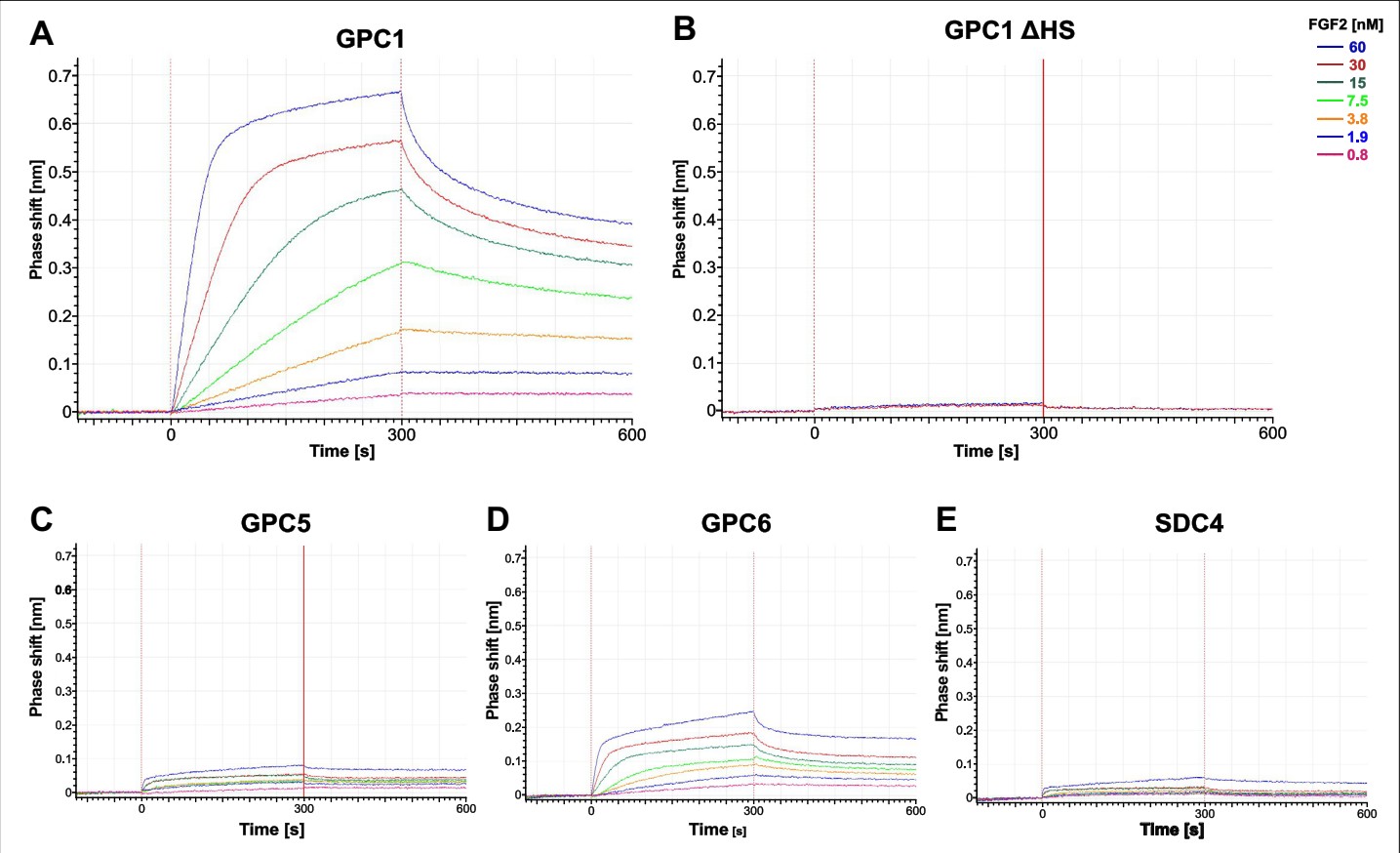

**Figure 6.** Glypican-1 (GPC1) and fibroblast growth factor 2 (FGF2) form a strong pair of interaction partners. Recombinant constructs encoding soluble ectodomains of GPC1 (panel A), GPC1 ΔHS (panel B; a mutant form to which heparan sulfate chains cannot be added), GPC5 (panel C), GPC6 (panel D), and SDC4 (panel E; a member of the syndecan family of heparan sulfate proteoglycans) were expressed and purified from HEK293 cells (see *Figure 6—figure supplement 1*). Using biolayer interferometry, interactions studies with temporal resolution visualizing both association and dissociation kinetics were conducted with purified FGF2 (*Figure 6—figure supplement 1*) at the concentrations indicated. The data shown are representative for two independent experiments. Experimental details are given in the Materials and methods section.

The online version of this article includes the following source data and figure supplement(s) for figure 6:

**Figure supplement 1.** Expression and purification of soluble recombinant forms of fibroblast growth factor 2 (FGF2) and the various heparan sulfate proteoglycans.

**Figure supplement 1—source data 1.** Unprocessed and uncropped image file of a SDS-PAGE analysis of the recombinant forms of GPC family members and fibroblast growth factor 2 (FGF2) used in the BLI experiments shown in Fig.

**Figure supplement 2.** Interaction studies between different kinds of heparan sulfate proteoglycans and growth factors or cytokines using biolayer interferometry.

receptors, and heparan sulfate chains. At both 10 and 1 ng/ml FGF2, all types of cells indicated showed reduced FGF2 signaling when treated with heparinases (*Figure 8A* [quantification with statistics] and *Figure 8B* [representative Western analysis of ERK phosphorylation]). By contrast, neither a knockout nor overexpression of GPC1 had any significant impact on phosphorylated ERK1/2 levels at both 10 and 1 ng/ml FGF2 added to cells (*Figure 8A, B*). These findings reveal a differential role of GPC1 in FGF2-related processes with GPC1 being essential for efficient secretion of FGF2 but being dispensable for the transmission of FGF2-dependent signals into cells.

## Discussion

In this study, we report on the surprising identification of the GPC family member GPC1 as an HSPG with a specialized function in driving unconventional secretion of FGF2. While we found that the total amounts of glycosaminoglycans including heparan sulfate chains are not significantly altered in GPC1

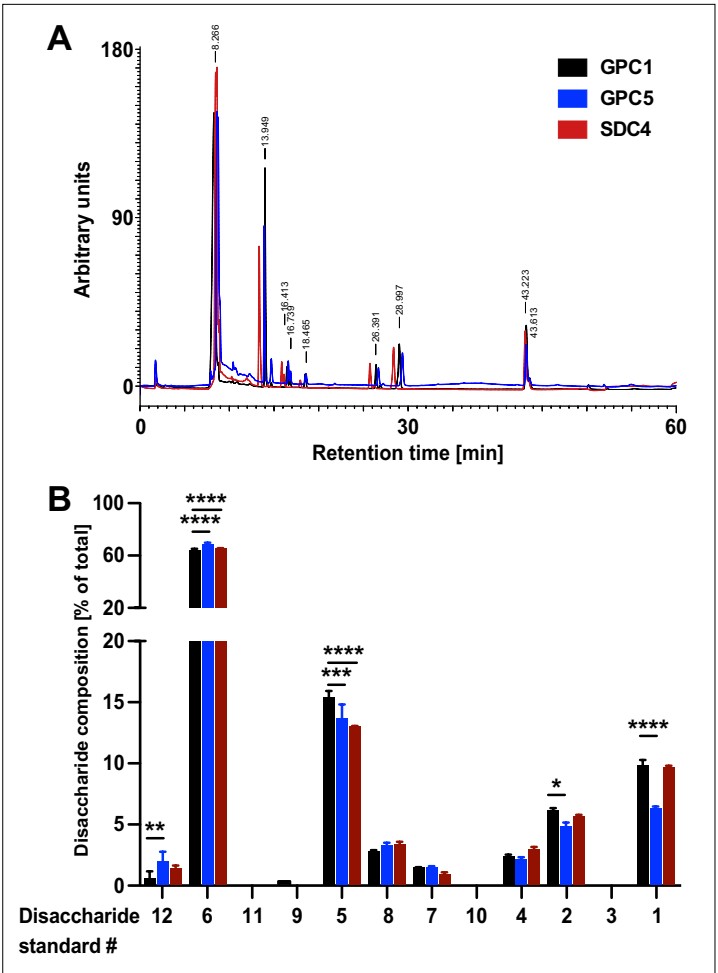

**Figure 7.** Quantitative characterization of the disaccharide contents of heparan sulfate chains derived from Glypican-1 (GPC1), GPC5, and SDC4. The recombinant forms of GPC1, GPC5, and SDC4 described in *Figure 6* and *Figure 6—figure supplement 1* were treated with a mixture of the heparinases I + II and III to release the disaccharide units of their heparan sulfate chains. These were then identified by their retention times in a analytical anion exchange HPLC setup using synthetic disaccharides as standards (*Figure 7—figure supplement 1*). For details, see Materials and methods. (**A**) Representative elution profile of the disaccharide units derived from the heparan sulfate chains of GPC1 (black), GPC5 (blue), and SDC4 (red). (**B**) Statistical analysis of four independent experiments providing the relative abundances of heparan sulfate disaccharide units corresponding to the 12 standards (*Figure 7—figure supplement 1*) contained in GPC1 (black), GPC5 (blue), and SDC4 (red). Standard deviations are shown. Statistics were based on a two-way ANOVA test combined with a Bonferroni post-test (*, p ≤ 0.05; **, p ≤ 0.01; ***, p ≤ 0.001, and ****, p ≤ 0.0001).

The online version of this article includes the following source data and figure supplement(s) for figure 7:

**Source data 1.** Raw data of the analytical HPLC experiments quantifying disaccharide units of the heparan sulfate chains of the proteoglycans indicated.

**Figure supplement 1.** Characterization of synthetic dissacharide standards corresponding to the building blocks of heparan sulfate chains using an analytical HPLC analysis.

knockout cells, we observed a pronounced decrease in FGF2 secretion efficiencies in the absence of GPC1. Overexpression of GPC5 (the second GPC family member expressed alongside GPC1 in HeLa cells) or SDC4, a member of the SDC family of HSPGs, did not rescue this process in a GPC1 knockout background. By contrast, overexpression of GPC1 in GPC1 knockout cells not only restored but rather caused a substantial increase of FGF2 secretion efficiencies. Therefore, GPC1 is a rate-limiting component of the FGF2 secretion machinery that is required for efficient transport of FGF2 into the extracellular space. These findings have implications for the molecular mechanism by which

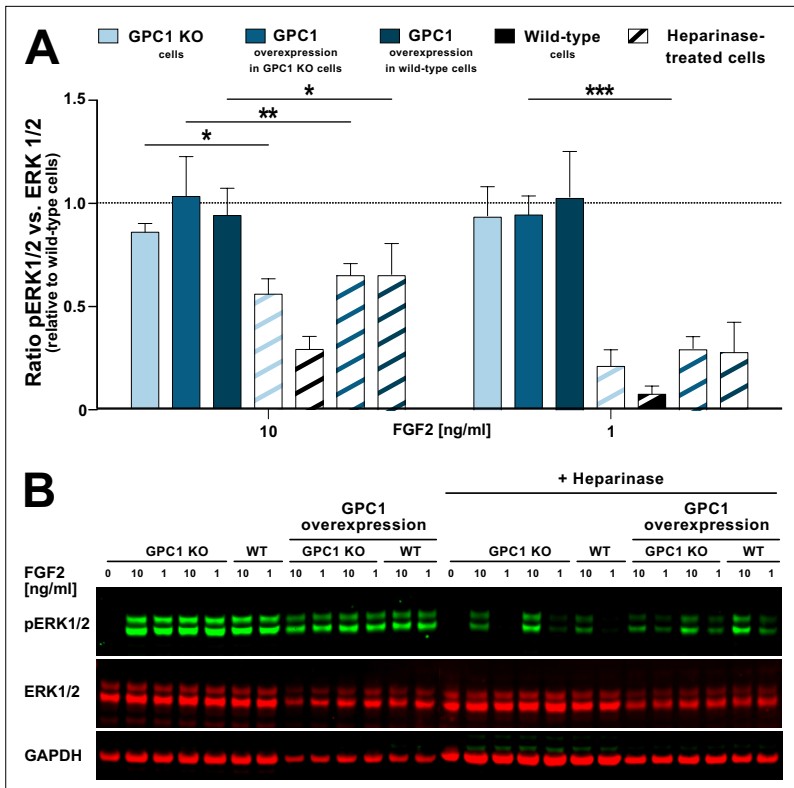

**Figure 8.** Glypican-1 (GPC1) is dispensable for fibroblast growth factor 2 (FGF2) signaling. Various forms of HeLa cells including wild-type and GPC1 knockout cells as well as cells overexpressing GPC1 in either a wild-type or a GPC1 knockout background were treated with recombinant FGF2 at the concentrations indicated. Where indicated, cells were treated with a mixture of heparinase I, II, and III used as a positive control. As a read-out for FGF signaling, the ratio between phosphorylated and unphosphorylated ERK1/2 was determined. For experimental details, see Materials and methods. (**A**) Quantitative analysis of the pERK1/2 to ERK1/2 ratio (n=4; standard deviations are shown). The statistical analysis was based on a one-way ANOVA test combined with Tukey's post hoc test (*, $p \leq 0.05$; **, $p \leq 0.01$; and ***, $p \leq 0.001$). (**B**) A representative Western analysis was used for the quantification and statistical analysis shown in panel **A**. The GAPDH signal was used as a loading control.

The online version of this article includes the following source data for figure 8:

**Source data 1.** Raw data of the cell-based signaling experiments quantifying fibroblast growth factor 2 (FGF2)-induced signaling cascades under the conditions indicated.

GPC1 is functioning in this process. Beyond the role of cell surface heparan sulfate chains in capturing and disassembling FGF2 oligomers at the outer plasma membrane leaflet (*Zehe et al., 2006*; *Nickel, 2007*; *Rabouille, 2017*; *Dimou and Nickel, 2018*; *Pallotta and Nickel, 2020*), they further suggest that GPC1 is already required for membrane insertion of FGF2 oligomers, an upstream step that is initiated at the inner plasma membrane leaflet. Based upon the unique positioning of the heparan sulfate chains of GPC1 in close proximity to its GPI anchor and therefore the membrane surface (*Prydz and Dalen, 2000*; *Blackhall et al., 2001*; *Nakato and Kimata, 2002*; *De Pasquale and Pavone, 2020*), the FGF2 binding sites in GPC1 appear to be required for stabilizing the first subunits of membrane-spanning FGF2 oligomers as they surface at the outer plasma membrane leaflet. In this way, the likeliness of these intermediates to be formed could be strongly increased when GPC1 is overexpressed. This, in turn, would result in FGF2 membrane translocation events occurring more frequently. These observations are consistent with previous findings demonstrating that, compared to wild-type cells, higher forms of membrane-inserted FGF2 oligomers do not accumulate in cells lacking cell surface heparan sulfates (*Dimou et al., 2019*).

To obtain insight into the molecular mechanism by which GPC1 drives unconventional secretion of FGF2, we conducted quantitative binding studies of FGF2 with GPC1 and other HSPGs using recombinant components purified from HEK cells. These experiments revealed GPC1 to be the strongest

FGF2 binding partner with the heparan sulfate chains being essential for this interaction. Based on a quantitative analysis of the disaccharide subunits in the heparan sulfate chains of GPC1, GPC5, and SDC4, we found disaccharides with N-linked sulfate groups corresponding to the standards 1, 2, and 5 enriched in GPC1 over GPC5 and SDC4. In particular, a tri-sulfated disaccharide (corresponding to standard 1 in *Figure 7* and *Figure 7—figure supplement 1*) was found enriched in GPC1 over GPC5. By contrast, disaccharides lacking O- or N-linked sulfates (such as the ones corresponding to standards 6 and 12) were found enriched in GPC5 and SDC4 compared to GPC1. Consistently, a trimer of the tri-sulfated disaccharide enriched in GPC1 (corresponding to standard 1) has been found to be a strong binding motif for FGF2 in structural in vitro studies using synthetic heparin molecules (*Raman et al., 2003*). However, the differences in the overall abundance of the various disaccharide units found in GPC1 versus GPC5 versus SDC4 alone could not explain the dramatic differences regarding their affinity toward FGF2. Therefore, we propose the heparan sulfate chains of GPC1 to contain multiple copies in a clustered manner of the FGF2 hexasaccharide ligands containing three tri-sulfated disaccharides. Such an arrangement would produce a high avidity and, therefore, a strong affinity of GPC1 toward FGF2. While it remains a goal for future studies to identify the precise structure of the FGF2 binding sites in the heparan sulfate chains of GPC1, our interpretation of the current findings provides a plausible explanation for the observed differences regarding the FGF2 binding efficiencies toward GPC1, GPC5, and SDC4. Along with the unique spatial organization of the heparan sulfate chains of GPC1 being arranged in close proximity to the plasma membrane surface (*Prydz and Dalen, 2000*; *Blackhall et al., 2001*; *Nakato and Kimata, 2002*; *De Pasquale and Pavone, 2020*), the identification of GPC1 being a high-affinity binding partner of FGF2 provides insights explaining its prominent role in driving unconventional secretion of FGF2 from cells.

Based on the results presented in this study, one may wonder whether GPC1 is also relevant for unconventionally secreted proteins other than FGF2. While a comprehensive analysis looking into this aspect will be a goal of future studies, we tested galectin-1 and galectin-3 as potential binding partners of GPC1. They belong to a family of lectins that are known to be secreted by unconventional means (*Rabouille, 2017*; *Dimou and Nickel, 2018*; *Popa et al., 2018*). In contrast to FGF2, despite being lectins as well, both galectin-1 and galectin-3 were incapable of interacting with GPC1. These findings suggest that GPC1 is not a general component of unconventional secretory processes but may represent a highly specific molecular component driving FGF2 secretion.

Another intriguing finding of this study was the observation that GPC1 is dispensable for FGF signaling. This was evident from experiments demonstrating that FGF2-induced signaling leads to similar levels of phosphorylated ERK1/2 levels in GPC1 knockout versus wild-type cells. Likewise, overexpression of GPC1 did not affect FGF2-induced activation of ERK1/2. These findings suggest that other HSPGs such as SDCs are sufficient to support FGF2 signaling. By contrast, unconventional secretion of FGF2 largely depends on the presence of GPC1 as the key factor that determines the efficiency of this process. With these observations, while GPC1 is not essential for FGF signaling, our study reveals an intimate relationship between FGF2 and GPC1 in the secretion of FGF2 from cells. Since both FGF2 and GPC1 are key components of tumor progression for a wide range of cancer types (*Akl et al., 2016*; *Pan and Ho, 2021*), we propose that the prominent function of GPC1 in driving efficient unconventional secretion of FGF2 might play a key role for tumor development such as acute myeloid leukemia (*Traer et al., 2016*; *Javidi-Sharifi et al., 2019*).

## Materials and methods

### Cell culture

Hela S3 cells were cultured in DMEM, supplemented with 10% FCS and 100 IU/ml penicillin and 100 μg/ml streptomycin at 37°C with 95% humidity and 5% $CO_2$. Human embryonic kidney EcoPack 2-293 cells (Clontech) were cultivated on collagen-coated (Collagen R; Serva Electrophoresis) plates under the same conditions. HEK293 cells were cultured under the same conditions. For protein purification, HEK293 cells were grown in EX-CELL ACF CHO Medium (Sigma-Aldrich, C5467) supplemented with 100 IU/ml penicillin and 100 μg/ml streptomycin at 37°C with 95% humidity and 5% $CO_2$. CHO K1 cells were cultured in α-MEM medium supplemented with 10% FCS, 2 mM glutamine, 100 U/ml penicillin, and 100 μg/ml streptomycin at 37°C with 95% humidity and 5% $CO_2$. All cell lines used in this study were received from the Leibniz Institute DSMZ (German collection of microorganisms and

cell cultures GmbH). For human cell lines, their identities were confirmed by STR profiling. For CHO cells, identity and purity were analyzed by a multiplex cell contamination test (*Schmitt and Pawlita, 2009*). All cell lines tested negative for mycoplasma contaminations.

## Generation of stable cell lines

For all experiments, stable cell lines were generated with a retroviral transduction system based on Moloney Murine Leukemia Virus as previously described (*Engling et al., 2002*). Virus production was performed in HEK293 cells with a stably integrated pVPack-Eco packaging system in its genome as well as the retroviral packaging proteins (EcoPack 2-293 cells). Proteins expressed upon induction with doxycycline, like FGF2-GFP, were cloned into the pRevTre2 vector, containing a Tet-response element and GPCs and SDCs were cloned into the pFB NEO vector. Retrovirus production was performed according to the MBS Mammalian Transfection Kit (Agilent Technologies) and virus was harvested after 2 days from confluent cells. Hela S3 and CHO K1 cells constitutively expressing the murine cationic amino acid transporter MCAT-1 (*Albritton et al., 1989*) and a Tet-On transactivator, rtTA2-M2 (*Urlinger et al., 2000*), were transduced with the freshly harvested virus. GFP-expressing cells were selected by FACS, untagged protein containing cells were selected with G418 and protein expression levels were analyzed by Western blot. HSPGs were detected after Heparinase III digest (NEB) using a monoclonal antibody (3G10) direct against the glycosylation attachment site in the core protein structure (370260, abcam).

## Generation of knockout cell lines

Knockout cells were generated via CRISPR-Cas9 as previously described (*Ran et al., 2013*). Briefly, gRNAs for GPC1 (exon 2: fwd 5'- CACCGTGCAGCAGGTGTAGCCCTG-3'; rev 5'- AAACCAGG GCTACACCTGCTGCAC-3') and GPC5 (exon 3: fwd 5'-CACCGATACTCAGAATGCATCCGGA-3'; rev 5'-AAACTCCGGATGCATTCTGAGTATC-3') were subcloned into pSpCas9(BB)-2A-RFP (based on pSpCas9(BB)-2A-GFP (PX458); *Ran et al., 2013* using BbsI [NEB #R3539]). GPC1, GPC5, and GPC1/5 knockouts were generated in HeLa S3 FGF2-GFP cells that were grown in six-well plates to 80% confluency. Cells were transfected with 2 µg DNA using FuGENE HD Transfection Reagent (REF E2311, Promega). After 24 hr, cells were transferred to a 10 cm dish and FGF2-GFP expression was induced via addition of doxycycline (1 µM). Forty-eight hr after transfection, single cell clones were sorted for GFP and RFP fluorescence into 96-well plates. Clones were validated for GPC1/5 knockouts via Western analysis following heparinase III digestion using a monoclonal antibody (3G10) direct against the glycosylation attachment site in the core protein structure (370260, abcam) or via sequencing (GPC1: 5'-ACTCACCATCGAAGCTG-3' and GPC5: 5'-GCGGCTGGGCAGCAGGGACCT -3') as indicated.

## Identification of proteins in proximity to FGF2 in cells

Cells were detached with Gibco Cell Dissociation Buffer (Thermo Fisher Scientific) and equal cell numbers were lysed in 2 ml Cyto0.2 Buffer (40 mM HEPES pH7.4, 120 mM KCl, 2 mM EGTA, 0.4% glycerol, 0.2% NP-40, protease inhibitors) for 30 min at 4°C while rotating. Nuclear fraction was pelleted for 3 min at 1000× *g*, 4°C and the cytosolic fraction was supplemented with 0.4% SDS. After washing the pellet with Cyto0.1 Buffer (40 mM HEPES pH 7.4, 120 mM KCl, 2 mM EGTA, 0.4% glycerol, 0.1% NP-40, protease inhibitors) and Cyto0.0 Buffer (40 mM HEPES pH 7.4, 120 mM KCl, 2 mM EGTA, 0.4% glycerol, 0.0% NP-40, protease inhibitors), the nuclear fraction was lysed with Lysis Buffer (50 mM Tris pH 7.4, 500 mM NaCl, 0.4% SDS, 5 mM EDTA). Both fractions were sonicated four times for 30 s at 4°C and centrifuged for 10 min at 16,000× *g* to remove debris.

Hela S3 cells stably expressing myc-tagged BirA* or myc-tagged FGF2-BirA* under a doxycycline-inducible promoter were supplemented with doxycycline (1 µM) for 48 hr and biotin (50 µM) during the last 36 hr of culture. Cytosolic fractions generated as described above were adjusted to 2% Triton X-100 and 150 mM NaCl using 50 mM Tris pH 7.4 before incubation with 200 µl streptavidin-coupled Dynabeads (M-280, 6.7 × 108 beads/ml, Invitrogen) overnight at 4°C while rotating. Beads were washed two times for 8 min while rotating with each buffer: W1 (2% SDS); W2 (0.1% sodium deoxycholate, 1% Triton X-100, 500 mM NaCl, 1 mM EDTA, 50 mM HEPES, pH 7.4); W3 (10 mM Tris pH 8, 250 mM LiCl, 1 mM EDTA, 0.5% NP-40, 0.5% sodium deoxycholate); and W4 (50 mM Tris pH 7.4, 50 mM NaCl, 0.1% NP-40). Proteins were eluted in 4× sample buffer (40% glycerol, 240 mM Tris-HCl

pH 6.8, 8% SDS, 5% β-mercaptoethanol, and bromophenol blue) for 15 min at 95°C and separated in a 1.5 mm 10-well 4–12% pre-cast gradient gel. After gels were washed three times with ddH$_2$O for 10 min, a Colloidal Coomassie staining (0.02% CBBG-250, 5% aluminum sulfate (14,18)-hydrate, 10% EtOH, 6.8% orthophosphoric acid) was performed overnight at room temperature (RT) and afterward destained (10% EtOH, 1.7% orthophosphoric acid) for 1 hr. Gels were washed twice with ddH$_2$O for 10 min before bands were cut and sent to FingerPrints Proteomics (Dundee University, Scotland).

Gel pieces were subjected to in-gel reductive alkylation and trypsin digest. Peptides were separated using strong cation exchange (SCX) fractionation before 1D nano-LC-MS/MS of each SCX fraction. The resultant mass spectrometry data from each fraction was merged prior to Mascot database search using the MaxQuant software (V1.5.5.1) with the human Uniprot sequence database (UP000005640, 9606 – *Homo sapiens*, 26.08.2015) for protein identification. Digestion mode was set to specific Trypsin/P with maximal two missed cleavages. Carbamidomethyl was set as fixed modification, acetylation of the N-terminus, oxidation (M), and biotinylation at lysine were chosen as variable modifications. Protein quantification was performed with unique and razor peptides. Protein intensities from the resulting ProteinGroups.txt file of three independent biological replicates were analyzed using the Perseus software (V1.5.5.1). Only proteins detected in the FGF2-BirA sample of at least two out of three biological replicates were considered for the analysis. The fold change between the FGF2-BirA and BirA-group was calculated from the means and log$_2$-tansformed with standard imputation based on normal distribution. Significant differences between the groups were analyzed by two-sided t-test.

## FGF2 secretion experiments based on cell surface biotinylation

$3 \times 10^5$ cells were seeded 48 hr prior to biotinylation and incubated with 1 µg/ml doxycycline after 24 hr for induction of FGF2-GFP expression. For biotinylation, cells were placed on ice and washed twice with PBS-Ca/Mg (1 mM MgCl$_2$, 0.1 mM CaCl$_2$). Cells were incubated with 1 mg/ml sulfo-NHS-SS-biotin (Thermo Fisher Scientific, 21331) in incubation buffer (150 mM NaCl, 10 mM triethanolamine pH 9.0, and 2 mM CaCl$_2$) on ice for 30 min with shaking, subsequently washed once with quenching buffer (100 mM glycine in PBS-Ca/Mg) and quenched for 20 min while shaking. Cells were washed twice with PBS and lysed for 10 min in lysis buffer (62.5 mM EDTA pH 8.0, 50 mM Tris-HCl pH 7.5, 0.4% sodium deoxycholate, 1% NP-40, and protease inhibitors from Roche) at 37°C. Lysed cells were detached via scraping and transferred into an Eppendorf tube. Cells were sonicated 3 min in a sonification bath and incubated 15 min at RT with vortexing every 5 min to solubilize all proteins. Lysates were cleared via 10 min centrifugation at 13,000 rpm 4°C in a table-top centrifuge. Meanwhile, Pierce Streptavidin UltraLink Resin (Thermo Fisher Scientific, 53114) was washed twice in lysis buffer via 1 min centrifugation at 3000× *g*; 5% input was taken from cleared lysates, mixed 1:1 with 4× sample buffer (40% glycerol, 240 mM Tris-HCl pH 6.8, 8% SDS, 5% β-mercaptoethanol, and bromphenol blue) and boiled for 10 min at 95°C. The remaining lysate was incubated for 1 hr at RT with over-head turning. Beads were spun down and washed once with wash buffer 1 (0.5 M NaCl in lysis buffer) and thrice with wash buffer 2 (0.5 M NaCl in lysis buffer containing 0.1% NP-40) via centrifugation. Beads were eluted via boiling in 4× sample buffer at 95°C for 10 min.

## FGF2 secretion experiments based on flow cytometry

After induction with doxycycline (1 µg/ml) for 16 hr in six-well plates, cells were washed once with PBS and collected after treatment with PBS supplemented with 5 mM EDTA. Cell surfaces were stained with 300 µl complete medium containing anti-FGF2 antibody (1:100, *Engling et al., 2002*; *Zehe et al., 2006*) (at 4°C for 1 hr). After centrifugation for 10 min at 500× *g*, pellets were washed with PBS and resuspended in 100 µl complete medium containing anti-rabbit APC antibody (1:500) followed by incubation for 30 min in the shaker at 800 rpm. After a final wash with PBS, cells were recovered in 300 µl PBS for FACSCaliburFlow Cytometer (Becton Dickinson) measurement for detection of cell surface bound FGF2-GFP.

## Visualization of endocytosis comparing fluorescent forms of transferrin and recombinant FGF2

The HeLa S3 wild-type, GPC1 knockout, and GPC1 knockout + GPC1 cells used in *Figures 2 and 4* were cultivated in µ-Slide 8 Well Glass Bottom dishes in the absence of doxycycline. This prevented the expression of FGF2-GFP so that endocytosis experiments with purified FGF2-GFP and fluorescent

transferrin could be conducted without interference. For live cell imaging, cells were washed twice with cold Live Cell Imaging Solution (Thermo Fisher Scientific) and incubated for 5 min on ice. Cells were imaged with a Zeiss LSM 800 confocal microscope using a Zeiss Plan-APOCHROMAT 63×/ 1.4 Oil DIC objective. Imaging was started directly after replacing the solution with cold live cell imaging solution containing both 25 µg/ml Transferrin-Alexa Fluor 546 (Thermo Fisher Scientific) and 5 µg/ml recombinant FGF2-GFP (*Steringer et al., 2017*). Time-lapse videos were recorded with images being acquired every 10 s for a total of 20 min for each cell line. In parallel experiments, still images of fixed cells (4% PFA; Electron Microscopy Science) were taken at time points of up to 60 min using the same microscopy settings as indicated above. For time points 0, 5, and 10 min, different laser power and digital gain settings were used due to smaller fluorescent signals at shorter incubation times. Images and videos were processed using Fiji (*Schindelin et al., 2012*).

## Single particle TIRF translocation assay

Quantification of secreted FGF2-GFP particles was achieved employing a previously established single particle TIRF assay (*Dimou et al., 2019*). Wide-field fluorescence and TIRF images were acquired using an Olympus IX81 xCellence TIRF microscope equipped with an Olympus PLAPO x100/1.45 Oil DIC objective lens and a Hamamatsu ImagEM Enhanced (C9100-13) camera. Data were recorded and exported in Tagged Image File Format (TIFF) and analyzed via Fiji (*Schindelin et al., 2012*). For the quantification of FGF2-GFP translocation to cell surfaces, CHO K1 cells were seeded in µ-Slide 8 Well Glass Bottom plates (ibidi) followed by incubation for 24 hr in the presence of 1 µg/ml doxycycline to induce FGF2-GFP expression (for the experimental condition at high FGF2-GFP expression levels) or without doxycycline incubation (for the experimental condition at low FGF2-GFP expression levels). Following incubation, the medium was removed and cells were rinsed three times with Live Cell Imaging Solution (Thermo Fisher Scientific). Cells were further incubated on ice with membrane impermeable Alexa Fluor 647-labeled anti-GFP nanobodies (Chromotek) for 30 min. Afterward, they were rinsed three times with PBS and fixed with 4% PFA (Electron Microscopy Sciences) for 20 min at RT. GFP fluorescence was excited with an Olympus 488 nm, 100 mW diode laser. Nanobody fluorescence was excited with an Olympus 640 nm, 140 mW diode laser. The quantification of FGF2-GFP particles on cell surfaces was achieved through a quantitative analysis of TIRF images. The frame of each cell was selected by wide-field imaging. For the experimental condition at low FGF2-GFP expression levels, the EM Gain for the wide-field (GFP) was adjusted in order to properly select the cell area. The number of nanobody particles were normalized to the cell surface area ($\mu m^2$). The total number of nanobody particles per cell was quantified employing the Fiji plugin TrackMate (*Tinevez et al., 2017*). Background fluorescence was subtracted for all representative images shown.

## Quantification of FGF2 binding to cell surfaces

Cells grown to confluency were washed with PBS and detached by cell dissociation buffer. After cell counting, $2 \times 10^5$ cells were collected and washed again with PBS. Cell pellets were resuspended in 200 µl PBS and mixed with 200 µl of FGF2-GFP (5 µg) followed by 1 hr incubation on a rotating wheel at RT. Cells were washed once with PBS and the pellet was resuspended in 200 µl PBS before analysis using a FACSCalibur flow cytometer (Becton Dickinson) for GFP intensities. Intensity values were normalized to wild-type cell intensities.

## Quantification of cellular glycosaminoglycan chains

Assays were performed according to the manufacturer's Blyscan (biocolor) protocol. Briefly, cells were grown to confluency for 72 hr. After a PBS wash, cells were detached with PBS supplemented with 5 mM EDTA, and cells were dissolved in 400 µl Papain extraction buffer to be afterward incubated for 6 hr at 65°C on a shaker; 100 µl of sample were mixed with 1 ml Blyscan dye reagent for 30 min while shaking and precipitated GAGs were pelleted for 10 min at 12,000 rpm in a centrifuge. After removal of supernatant, the pellet was dissolved in 500 µl dissociation reagent for 10 min in shaker and 200 µl of supernatant were analyzed in 95-well plate at plate reader at 656 nm. Heparan sulfate chains were additionally analyzed by incubation of 100 µl of papain digested GAG sample with sodium nitrite (100 µl), followed by addition of 100 µl acetic acid. After vortexing, the samples were incubated for 60 min at RT and nitrous acid was removed by addition of 100 µl ammonium sulfamate reagent for

10 min; 100 μl of neutralized sample were analyzed as described above to quantify O-sulfated GAGS. Finally, to analyze the N-sulfated GAGs, results were subtracted from the total GAG amounts.

## Protein expression and purification

His-FGF2 (vector pQE30) and His-FGF2-GFP (vector pET15b) were purified from the *Escherichia coli* strains W3110Z1 and BL21 Star, respectively. Following o/n expression at 25°C, proteins were purified sequentially by Ni-NTA affinity chromatography (HisTrap FF, GE Healthcare), heparin chromatography (HiTrap Heparin HP, GE Healthcare), and size exclusion chromatography using a Superdex 75 column. Proteins were snap-frozen in aliquots and stored at –80°C. The GPCs indicated and SDC4 were cloned into the pcDNA3.1 vector containing a BM40 signal peptide (replacing of the original one) and a His-tag instead of a GPI anchor or a transmembrane domain (*Figure 6—figure supplement 1*). Proteins were expressed in HEK293 cells and supernatants were harvested after 4 days. Following centrifugation and filtering (0.2 μm), proteins were purified via Ni-NTA affinity chromatography followed by size exclusion chromatography using a Superose 6 and Superdex 200 column.

## Biolayer interferometry to quantify protein-protein interactions

The biolayer interferometry allows a label-free analysis of real-time interaction events due to an optical detection of biomolecules that bind to the fiber-optic biosensors. Upon immobilization of the ligand to the biosensor, a shift in the interference spectrum of the reflected light is induced and can be detected. As soon as the analyte binds the ligand, a further increase of the optical thickness on the biosensor surface is detected by the additional wavelength shift, which is then reported as the wavelength change (nm) over time (s). Measurements were performed on the OctetRed96e system (Sartorius) using the Streptavidin sensors (18-5019 SA, Sartorius). Data were evaluated with the Data Analysis HT 12.0 software (Sartorius). Proteins were measured in black 96-well plates (655209, Greiner) in 200 μl for all samples. Biosensors were hydrated for 10 min before measurements in Octet Buffer (PBS, 0.02% Tween and 0.1% BSA) to remove sucrose coverage. Measurements were conducted with the plates shaking at 1000 rpm. All assays were performed with SA biosensors and the ligands were biotinylated. Proteins were labeled with EZ-Link NHS-PEG4-Biotin (Thermo Scientific A39259) in a 1:1 ration at 37°C for 30 min. Proteins were separated from non-bound biotin by Zeba Spin Desalting Columns (Thermo Scientific 89882). Biotinylation does not interfere with the binding kinetics, as the interaction takes place at the heparan sulfate chains. A loading scout was performed to find the optimal amount of bound ligand to the biosensor surface. All kinetic experiments were performed with 6 μg/ml of biotinylated ligand loaded to the sensor for 10 min. If not stated otherwise in the figure legends, the assay setup was as follows: (i) baseline in Octet Buffer (2 min), (ii) load with ligand (6 μg/ml, 10 min), (iii) wash in octet buffer (1 min), (iv) baseline II in octet buffer (1 min), (v) association of FGF2 (60 μM dilution series, 1 or 5 min), (vi) dissociation in octet buffer (1 or 5 min), (vii) recovery in glycine pH 1.7 and octet buffer (3 × 5 s each).

## FGF2 signaling assays

Hela S3 cells with a FGF2 and GPC1 knockout background were induced with recombinant FGF2 (1 and 10 ng/ml) for 20 min. In parallel, Heparinase digests were performed for 2 hr at 37°C with a mixture of Heparinase I + II and III (3.5 munits/ml, NEB) in Heparinase digestion buffer (20 mM Tris, 100 mM NaCl, and 1.5 mM CaCl$_2$, pH 7.4) before cell signaling was induced by FGF2. Cells were lysed and analyzed by Western blot for ERK1/2 (4696, CST) and pERK1/2 (9101, CST) levels. GAPDH (AM4300, Invitrogen) was detected as loading control.

## Identification and quantification of heparan sulfate disaccharides

Heparan sulfate disaccharides of HSPGs were analyzed as described previously (*Carnachan and Hinkley, 2017*). Briefly, proteins (1 mg/ml) were digested in 200 μl digestion buffer (100 mM NaOAc, 2 mM CaOAc, pH 7.0) with 1.75 mIU of Heparinase I + II and III (NEB) for 16 hr at 30°C. Heparinases were inactivated at 95°C for 10 min and denatured proteins were pelleted at 16,000× *g* for 10 min; 100 μl of supernatant was analyzed by HPLC with a strong anion exchange column (ProPac PA1, Thermo Fisher) with an elution gradient of 2 M NaCl, pH 3.5. Disaccharides were detected at an absorbance maximum of 232 nm. A standard mixture of heparan sulfate disaccharides (20 μg/ml; Iduron, UK) was analyzed to identify heparan sulfate disaccharides.

## Acknowledgements

This work was supported by grants from the Deutsche Forschungsgemeinschaft (WN; SFB/TRR 83 and SFB/TRR 186). We are indebted to Julien Bethune (HAW Hamburg, Germany) who helped designing the BioID screen reported in this study. We would like to further acknowledge help from Holger Lorenz (ZMBH imaging facility) and Monika Langlotz (ZMBH FACS facility). We thank Marco Binder (DKFZ Heidelberg) for providing HEK293 cells used for the production of recombinant HSPGs and the FingerPrints mass spectrometry unit at Dundee University contributing the proteomics analysis contained in this study.

## Additional information

### Funding

| Funder | Grant reference number | Author |
|---|---|---|
| Deutsche Forschungsgemeinschaft | SFB/TRR 83 - A5 | Carola Sparn<br>Walter Nickel |
| Deutsche Forschungsgemeinschaft | SFB/TRR 186 - A1 | Eleni Dimou<br>Annalena Meyer<br>Roberto Saleppico<br>Helge Ewers<br>Walter Nickel |

The funders had no role in study design, data collection and interpretation, or the decision to submit the work for publication.

### Author contributions

Carola Sparn, Eleni Dimou, Annalena Meyer, Roberto Saleppico, Sabine Wegehingel, Matthias Gerstner, Severina Klaus, Data curation, Formal analysis, Investigation, Methodology; Helge Ewers, Conceptualization, Methodology, Supervision, Validation; Walter Nickel, Conceptualization, Funding acquisition, Supervision, Writing - original draft, Writing - review and editing

### Author ORCIDs

Eleni Dimou (iD) http://orcid.org/0000-0003-4885-135X
Helge Ewers (iD) http://orcid.org/0000-0003-3948-4332
Walter Nickel (iD) http://orcid.org/0000-0002-6496-8286

### Decision letter and Author response

Decision letter https://doi.org/10.7554/eLife.75545.sa1
Author response https://doi.org/10.7554/eLife.75545.sa2

## Additional files

### Supplementary files

- Transparent reporting form
- Source data 1. Raw data from *Figures 1–8*.

### Data availability

All data generated or analysed during this study are included in the manuscript and supporting file; source data files have been provided for various figures in a compressed zip file.

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
