## [Editor Report]

FGF2 moves directly from the cytoplasm through the plasma membrane in a reaction driven by its subsequent high affinity binding to cell surface heparan sulfate proteoglycans. This study, surprisingly, identifies Glypican-1 as the principal proteoglycan involved, possibly involving a unique tri-sulfated disaccharide binding site in close proximity to the cell surface. Thus, Glypican-1 is new component in the pathway of unconventional secretion of FGF2.

---

## [Decision Letter]

**Decision letter after peer review:**

Thank you for submitting your article "Glypican-1 drives unconventional secretion of Fibroblast Growth Factor 2" for consideration by *eLife*. Your article has been reviewed by 3 peer reviewers, one of whom is a member of our Board of Reviewing Editors, and the evaluation has been overseen by Suzanne Pfeffer as the Senior Editor. The reviewers have opted to remain anonymous.

Essential revisions:

This manuscript describes roles of the heparan sulfate glycosaminoglycan (GAG) glypican GPC1 in non-conventional secretion of FGF2. GPC1 was identified in a proteomic screen for proteins that interact with FGF2. GPC1 facilitates export of FGF2, acts as a highly specific surface receptor for FGF2, but is not required for FGF2-induced activation of ERK.

Overall, the reviewers found the study to be significant, clearly written and mostly convincing. Please submit a revised version that addresses the following. Items 1-3 are considered major.

1. FGF2 binding to GPC1 is glycan dependent, based on the loss of binding after deglycosylation. However, the proposed increased concentration of the hexasaccharide ligand relative to other HSPGs has not been sufficiently demonstrated. Assuming that the critical disaccharide is trimerized to make the ligand for FGF2 (Figure 6 and Raman 2003), then the difference in trimer concentration between GPC1 vs. SDC4 is 1.2^3^=1.7-fold, based on random assembly of the glycan chain. This concentration difference is not enough to cause the huge difference in apparent binding affinity shown in Figure 6. Nevertheless, based on the effects of deglycosylation, binding is clearly glycan dependent. Therefore, there must be another source of the affinity difference. Two possibilities come to mind. First, the interaction between FGF2 and GPC1 would be dependent on FGF2-glycan binding, but also includes other high affinity/high specificity sites. A second possibility is that the hexasaccharide ligand is clustered in GPC1 but not in other GAGs. That clustering would specifically raise the avidity and apparent affinity of the interaction. These issues should be addressed.

2. Figure 2 shows a strong reduction of FGF2 secretion upon GPC1 KO using the surface biotinylation assay (approx. 75% with little error). This is the critical evidence that GPC1 is required, yet in Figure 4A and D the result looks much less convincing (50% with large error). For Fig4D the WT data are present and it seems questionable whether there is a significant difference. This needs to be explained and/or corrected. Statistical significance of WT vs KO should be reported in Figure 1 and the result should be reproducible throughout, if using the same assay.

3. Is there a pool of FGF2 arrested at the plasma membrane in cells lacking GPC1? Is this pool attached to the cytoplasmic face of the plasma membrane? Some level of understanding of the step where the function of GPC1 in the overall translocation of FGF2 across the plasma membrane is essential. The authors have shown before that Tec kinase and Na, K-ATPase are required for FGF2 secretion. I presume these two proteins are up stream of GPC1 in FGF2 translocation pathway. Can this be tested experimentally?

*Reviewer #1 (Recommendations for the authors):*

FGF2 binding to GPC1 is glycan dependent, based on the loss of binding after deglycosylation. However, the proposed increased concentration of the hexasaccharide ligand relative to other HSPGs has not been sufficiently demonstrated. Assuming that the critical disaccharide is trimerized to make the ligand for FGF2 (Figure 6 and Raman 2003), then the difference in trimer concentration between GPC1 vs. SDC4 is 1.2^3^=1.7-fold, based on random assembly of the glycan chain. This concentration difference is not enough to cause the huge difference in apparent binding affinity shown in Figure 6. Nevertheless, based on the effects of deglycosylation, binding is clearly glycan dependent. Therefore, there must be another source of the affinity difference. Two possibilities come to mind. First, the interaction between FGF2 and GPC1 would be dependent on FGF2-glycan binding, but also includes other high affinity/high specificity sites. A second possibility is that the hexasaccharide ligand is clustered in GPC1 but not in other GAGs. That clustering would specifically raise the avidity and apparent affinity of the interaction. These issues should be addressed.

*Reviewer #3 (Recommendations for the authors):*

The data are interesting, but the authors are encouraged to address the following issues.

1. Is there a pool of FGF2 arrested at the plasma membrane in cells lacking GPC1? Is this pool attached to the cytoplasmic face of the plasma membrane? Some level of understanding of the step where the function of GPC1 in the overall translocation of FGF2 across the plasma membrane is essential. The authors have shown before that Tec kinase and Na, K-ATPase are required for FGF2 secretion. I presume these two proteins are up stream of GPC1 in FGF2 translocation pathway. Can this be tested experimentally?

2. Does protease dependent cleavage and removal of GPC1 affect FGF2 translocation?

3. What are the relative levels of GPC1 and GPC5 at the cell surface? is the binding specificities of FGF2 to GPC1 reported here simply a reflection of the relative abundance of GPC 1 and GPC5.

4. Is GPC1 involved in capture and retention of FGF2 at the exoplasmic surface or does it affect the rate of endocytosis. Is it possible that in the absence of GPC1, FGF2 is secreted but rapidly endocytosed resulting in less FGF2 in the extracellular space.

5. Finally, the authors have shown that IL-1ß and FGF2 are trafficked by the same pathway of direct translocation across the membrane. Is IL-1ß traffic dependent on GPC1?

---

## [Author Response]

1. FGF2 binding to GPC1 is glycan dependent, based on the loss of binding after deglycosylation. However, the proposed increased concentration of the hexasaccharide ligand relative to other HSPGs has not been sufficiently demonstrated. Assuming that the critical disaccharide is trimerized to make the ligand for FGF2 (Figure 6 and Raman 2003), then the difference in trimer concentration between GPC1 vs. SDC4 is 1.2^3^=1.7-fold, based on random assembly of the glycan chain. This concentration difference is not enough to cause the huge difference in apparent binding affinity shown in Figure 6. Nevertheless, based on the effects of deglycosylation, binding is clearly glycan dependent. Therefore, there must be another source of the affinity difference. Two possibilities come to mind. First, the interaction between FGF2 and GPC1 would be dependent on FGF2-glycan binding, but also includes other high affinity/high specificity sites. A second possibility is that the hexasaccharide ligand is clustered in GPC1 but not in other GAGs. That clustering would specifically raise the avidity and apparent affinity of the interaction. These issues should be addressed.

It is true that, compared with GPC5 and SDC4, a simple enrichment of the critical tri-sulfated disaccharide (Figure 7, Figure S5 and Raman et al., 2003) alone would not explain the high affinity of FGF2 towards the heparan sulfate chains of GPC1. Therefore, as proposed by Reviewer #1, a possible explanation would be that the corresponding high affinity FGF2 hexasaccharide ligand is contained in multiple copies that cluster in the GPC1 heparan sulfate chains of GPC1, resulting in high avidity and therefore strong apparent affinity towards FGF2. In our view, this is indeed the most likely explanation for the observed differences in the apparent affinities of FGF2 towards GPC1 versus GPC5 and SDC4 (Figure 6). Unfortunately, this hypothesis currently cannot be tested experimentally since there are no methodologies available to sequence heparan sulfate chains derived from different types of proteoglycans. It is also technically not possible to swap authentic heparan sulfate chains between different types of proteoglycans, so it is also difficult to test whether there is a second non-glycan FGF2 binding site in the core protein of GPC1 that, combined with the heparan sulfate chains, generates a high affinity binding site for FGF2, a second possible explanation put forward by Reviewer 1. Of note, however, a heparan sulfate binding deficient variant of FGF2 (KRK127/128/133QQQ) does not bind to GPC1 (see Figure S4). This observation suggests that the heparan sulfate chains of GPC1 provide the sole binding site for FGF2. Therefore, the first possibility put forward by Reviewer 1 is clearly the most likely scenario, a clustering of the critical hexasaccharide FGF2 binding motifs in the heparan sulfate chains of GPC1. In the revised manuscript, these aspects are now discussed in great detail providing a compelling explanation for the observed high affinity interaction between FGF2 and GPC1.

2. Figure 2 shows a strong reduction of FGF2 secretion upon GPC1 KO using the surface biotinylation assay (approx. 75% with little error). This is the critical evidence that GPC1 is required, yet in Figure 4A and D the result looks much less convincing (50% with large error). For Fig4D the WT data are present and it seems questionable whether there is a significant difference. This needs to be explained and/or corrected. Statistical significance of WT vs KO should be reported in Figure 1 and the result should be reproducible throughout, if using the same assay.

As correctly noted by Reviewer 2, the strength of FGF2 secretion phenotypes comparing wild-type and GPC1 knockout cells differed to some extent between the series of experiments shown in Figure 2 versus those shown in Figure 4. As requested by Reviewer 2, in the revised versions of Figures 2 and 4, we have now indicated the statistical analysis for all conditions. While it is true that the absolute reduction in FGF2 secretion efficiencies is not the same, for all experiments done as part of this study, a GPC1 knockout always caused a significant reduction of FGF2 secretion efficiencies. In the redesigned Figure 4, there is an additional experimental condition that emphasizes this point, the reduction of FGF2 secretion in a GPC1 knock-out background when SDC4 is overexpressed. Again, a highly significant phenotype is observed with SDC4 overexpression not being able to rescue a GPC1 knock-out. So, while there is a certain degree of variation in the absolute reduction of FGF2 secretion rates in different series of experiments, the observed phenotypes are always statistically significant. At the same time, overexpression of GPC1 always causes a strong increase in FGF2 secretion efficiencies in a statistically significant manner. Therefore, even though there are certain variations between different series of experiments that were conducted independently by different authors of this manuscript, the described phenotypes are consistent and statistically significant in every single case. They fully support our conclusions with GPC1 being the predominant heparan sulfate proteoglycan driving efficient secretion of FGF2. This conclusion is further corroborated by our findings demonstrating GPC1 to have a much higher apparent affinity toward FGF2 compared to GPC5 and SDC4. In the revised manuscript, along with the new versions of Figure 2 and 4, we improved the discussion of these findings to make them more accessible to a broad readership.

3. Is there a pool of FGF2 arrested at the plasma membrane in cells lacking GPC1? Is this pool attached to the cytoplasmic face of the plasma membrane? Some level of understanding of the step where the function of GPC1 in the overall translocation of FGF2 across the plasma membrane is essential. The authors have shown before that Tec kinase and Na, K-ATPase are required for FGF2 secretion. I presume these two proteins are up stream of GPC1 in FGF2 translocation pathway. Can this be tested experimentally?

Regarding the mechanism of FGF2 membrane translocation, Reviewer 3 raised an interesting aspect about the sequence of events along with the spatio-temporal organization of the Na,K-ATPase, Tec kinase, PI(4,5)P_2_ and GPC1. As summarized in recent review articles on the topic (Dimou and Nickel 2018, Curr Biol; Pallotta and Nickel 2020, J Cell Sci), various lines of experimental evidence suggest the Na,K-ATPase, Tec kinase and PI(4,5)P_2_ (in this order) to be upstream of GPC1. First, a previous study from our lab has shown that the Na,K-ATPase acts upstream of PI(4,5)P_2_, most likely forming the first contact of FGF2 at the inner plasma membrane leaflet (Legrand et al., 2020, Commun Biol). This could be demonstrated by various kinds of FGF2 mutants with binding defects towards either the Na,K-ATPase or PI(4,5)P_2_. These experiments demonstrated that, in a cellular context, an FGF2 mutant that cannot bind to the Na,K-ATPase also fails to bind to PI(4,5)P_2_. The other way around, an FGF2 mutant that cannot bind PI(4,5)P_2_ was found to still bind to the Na,K-ATPase. These experiments were done in living cells using single molecule TIRF microscopy to quantify FGF2 recruitment at the inner plasma membrane leaflet (Legrand et al. ,2020, Commun Biol). Likewise, Tec kinase was also shown to act upstream of PI(4,5)P_2_ because Tec kinase-mediated phosphorylation of FGF2 enhances FGF2 oligomerization and membrane pore formation (Steringer et al., 2012, J Biol Chem). Regarding the role of heparan sulfate chains, in previous studies, we compared wild-type cells with heparan sulfate deficient cells. These studies demonstrated that FGF2 oligomers do not accumulate at the inner plasma membrane leaflet when heparan sulfate chains including those from GPC1 are absent from the cell surface (Dimou et al., 2019, J Cell Biol). Finally, we reconstituted FGF2 membrane translocation with purified components demonstrating that PI(4,5)P_2_ and heparan sulfate chains need to be on opposing sides of the membrane to promote FGF2 membrane translocation (Steringer et al., 2017, *eLife*). This shows that GPC1 is acting downstream of the other components which is consistent with its localization on cell surfaces and the transport direction of FGF2 across the plasma membrane.

The findings from previous studies match our interpretations from the current manuscript. Since we demonstrate GPC1 overexpression to cause the rate of FGF2 secretion to go up strongly (Figures 2 and 4), a plausible explanation would be that GPC1 is not only capturing and disassembling FGF2 oligomers at the outer leaflet (Steringer et al., 2017, *eLife*; Dimou et al., 2019, J Cell Biol) but rather is already required in a preceding step facilitating membrane insertion of FGF2 oligomers. An interesting model would be that, at the inner plasma membrane leaflet, the likeliness of FGF2 to assemble into membrane-spanning oligomers in a PI(4,5)P_2_-dependent manner is increased when a binding site is available at the outer leaflet. We believe this binding site for FGF2 is produced by the high affinity heparan sulfate chains of GPC1. This mode of action provides a compelling explanation of how GPC1 can act as the rate-limiting factor of the unconventional secretory pathway of FGF2. This hypothesis is put forward in detail in the first paragraph of the discussion of our manuscript.